# Inhibitory basal ganglia nuclei differentially innervate pedunculopontine nucleus subpopulations and evoke differential motor and valence behaviors

**Michel Fallah[1,2]\*, Kenea C Udobi[2], Aleksandra E Swiatek[2], Chelsea B Scott[2], Rebekah C Evans[1,2]\***

[1]Interdisciplinary Program in Neuroscience, Georgetown University, Washington DC, United States; [2]Department of Neuroscience, Georgetown University Medical Center, Washington DC, United States

**\*For correspondence:**
mf1278@georgetown.edu (MF);
re285@georgetown.edu (RCE)

**Competing interest:** The authors declare that no competing interests exist.

## eLife Assessment

Fallah et al carefully dissect projections from substantia nigra pars reticulata (SNr) and the globus pallidus externa (GPe) - two key basal ganglia nuclei - to the pedunculopontine nucleus (PPN), a brainstem nucleus that has a central role in motor control. They consider inputs from these two areas onto three types of downstream PPN neurons - GABAergic, glutamatergic, and cholinergic neurons - and carefully map connectivity along the rostrocaudal axis of the PPN. Overall, this **important** study provides **convincing** data on PPN connectivity with two key input structures that will provide a basis for further understanding PPN function.

**Abstract** The canonical basal ganglia model predicts that the substantia nigra *pars reticulata* (SNr) will inhibit locomotion and the globus pallidus *externa* (GPe) will enhance it. In mice, we use in vivo optogenetics to show that the GPe exerts non-canonical effects on locomotion while the SNr has no gross motor impact through inhibition of the pedunculopontine nucleus (PPN). We show that these structures mediate opposing effects on reward and that activation of *substantia nigra pars compacta* (SNc) dopaminergic axons in the PPN is rewarding. We use ex vivo whole-cell recording with optogenetics in mice to comprehensively dissect SNr and GPe synaptic connections to regionally- and molecularly-defined populations of PPN neurons. The SNr inhibits all PPN subtypes but most strongly inhibits caudal glutamatergic neurons. The GPe selectively inhibits caudal glutamatergic and GABAergic neurons, avoiding both cholinergic and rostral cells. This circuit characterization reveals non-canonical basal ganglia pathways for locomotion and valence.

## Introduction

The pedunculopontine nucleus (PPN) is a brainstem structure heavily interconnected with subcortical structures, such as the basal ganglia, thalamus, and spinal cord. The PPN has been implicated in locomotor control and valence processing (*Ryczko, 2024*; *Bastos-Gonçalves et al., 2024*; *Mena-Segovia and Bolam, 2017*; *French and Muthusamy, 2018*). However, understanding PPN circuitry is complicated by its anatomical and molecular heterogeneity. It has three molecularly defined cell types, distinguished by their major neurotransmitter (*Wang and Morales, 2009*) and distinct rostral and caudal topography. While cholinergic and glutamatergic neurons are more densely packed in the caudal PPN, GABAergic neurons are more densely packed in the rostral PPN (*Wang and Morales,*

*2009*; *Mena-Segovia et al., 2009*). These molecularly defined populations display very minimal overlap (*Wang and Morales, 2009*; *Yoo et al., 2017*; *Steinkellner et al., 2019*). This unique topography has been conserved across species (*Martinez-Gonzalez et al., 2011*; *Alam et al., 2011*; *Marín et al., 1998*), but it is unknown how the basal ganglia connects to each of the distinct cell types in the rostral and caudal PPN regions to modulate locomotor and valence outputs.

While the PPN interacts with all basal ganglia nuclei (*Mena-Segovia and Bolam, 2017*; *Martinez-Gonzalez et al., 2011*), the inhibitory inputs from the substantia nigra *pars reticulata* (SNr) and the globus pallidus *externus* (GPe) to the PPN are of particular interest because of their roles in the canonical motor output pathways for 'stop' and 'go', respectively. Both the SNr and GPe are known to send axonal projections to the PPN (*McElvain et al., 2021*; *Zhao et al., 2022*; *Roseberry et al., 2016*; *Huerta-Ocampo et al., 2021*; *Dautan et al., 2021*; *Caggiano et al., 2018*; *Henrich et al., 2020*). Previous studies have identified SNr and GPe monosynaptic connections to the PPN cholinergic and glutamatergic neurons using rabies tracing (*Zhao et al., 2022*; *Roseberry et al., 2016*; *Huerta-Ocampo et al., 2021*; *Dautan et al., 2021*; *Caggiano et al., 2018*). However, inputs to the GABAergic neurons have only been described in terms of the entire mesencephalic locomotor region (MLR), which extends beyond the PPN (*Roseberry et al., 2016*). While SNr projections to MLR GABAergic neurons were identified but cannot be isolated to the PPN, GPe projections to the MLR GABAergic neurons were not detected (*Roseberry et al., 2016*). Of note, these previous studies consider the PPN as a whole rather than separating its rostral and caudal subregions. While viral tracing can provide insight into neuroanatomical connections, limitations of these techniques (*Ugolini, 2011*; *Linders et al., 2022*; *Glasgow et al., 2019*) and evidence that the rostral and caudal PPN may have distinct functions (*Goñi-Erro et al., 2023*; *Masini and Kiehn, 2022*; *Gut et al., 2022*; *Huang et al., 2024*) prompt a need for electrophysiological studies to determine whether the SNr and GPe inputs differentially innervate regionally defined PPN neurons.

Optogenetic manipulations of the molecularly defined PPN cell types have resulted in variable, and sometimes contradictory, locomotor and valence behaviors (*Yoo et al., 2017*; *Roseberry et al., 2016*; *Dautan et al., 2021*; *Caggiano et al., 2018*; *Goñi-Erro et al., 2023*; *Masini and Kiehn, 2022*; *Gut et al., 2022*; *Huang et al., 2024*; *Josset et al., 2018*; *Tello et al., 2024*; *Xiao et al., 2016*; *Dautan et al., 2016*; *Ferreira-Pinto et al., 2021*; *Zhang et al., 2024*; *Hormigo et al., 2019*; *Hormigo et al., 2021*). More recent studies have refined our understanding of these opposing behavioral effects by selectively stimulating the rostral or caudal GABAergic and glutamatergic PPN neurons (*Goñi-Erro et al., 2023*; *Masini and Kiehn, 2022*; *Gut et al., 2022*; *Huang et al., 2024*). In the basal ganglia, direct stimulation of molecularly defined SNr and GPe subpopulations can also have distinct effects on locomotion and valence (*Rizzi and Tan, 2019*; *Liu et al., 2023*; *Galaj et al., 2020*; *Mastro et al., 2014*; *Tian et al., 2018*; *Pamukcu et al., 2020*; *Cui et al., 2021*; *Aristieta et al., 2021*; *Isett et al., 2023*). Despite the potential for the basal ganglia to modify behavior through the motor brainstem (*Roseberry et al., 2016*), it is not clear how the SNr and GPe influence the rostral and caudal PPN to modify locomotion and valence.

Here, we use ex vivo whole-cell patch clamp electrophysiology paired with optogenetics to selectively stimulate the SNr or GPe axons while recording inhibitory inputs to the cholinergic, GABAergic, and glutamatergic PPN neurons to comprehensively characterize the strength and pattern of inhibition across the rostrocaudal PPN axis. We find that the GPe preferentially inhibits caudal GABAergic and glutamatergic PPN neurons, whereas the SNr most strongly inhibits a caudal 'hotspot' of PPN glutamatergic neurons. Stimulating SNr or GPe axons over the PPN in vivo evokes opposing valence processing outcomes with place aversion during SNr stimulation and place preference during GPe stimulation. Surprisingly, and counter to the predictions of the canonical basal ganglia model, the GPe decreases locomotion through its projections to the PPN. While both the SNr and GPe inhibit the PPN, our results show that each nucleus differentially modulates the activity of specific cell types in the rostral and caudal PPN and is implicated in non-canonical basal ganglia circuits for modulating locomotion and valence processing.

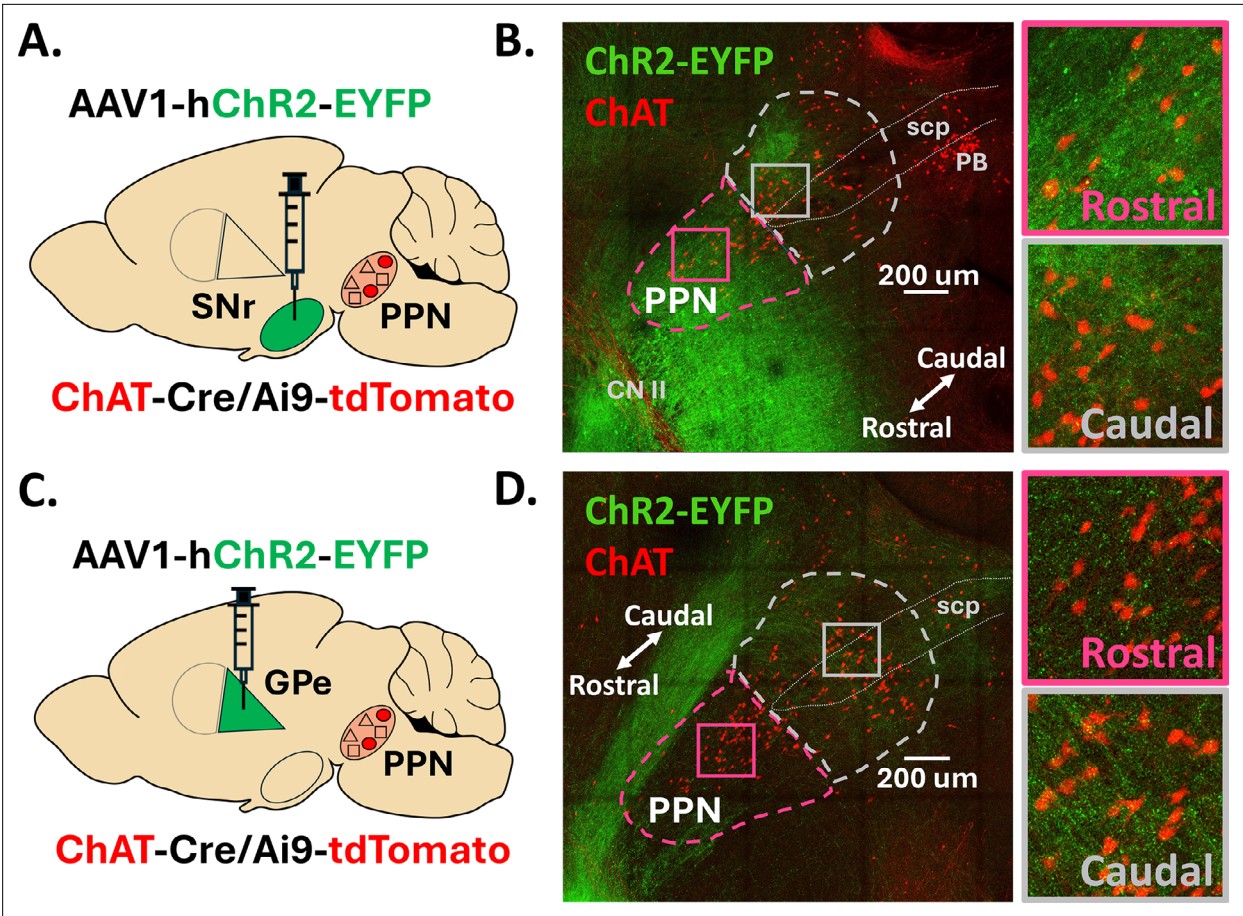

**Figure 1.** SNr and GPe axons display distinct distribution patterns across the rostral and caudal PPN. (**A,C**) Stereotaxic injection of AAV1 delivering *hSyn-ChR2-eYFP* to the SNr or GPe of ChAT-Cre/Ai9-tdTomato mice, respectively. (**B, D**) Confocal images of EYFP-filled SNr or GPe axons across the PPN, respectively. *CNII: cranial nerve II, scp: superior cerebellar peduncle; PB: parabrachial nucleus.*

The online version of this article includes the following figure supplement(s) for figure 1:

**Figure supplement 1.** Injection site verification.

## Results

### SNr and GPe axons display distinct distribution patterns across the rostral and caudal PPN

To characterize the axon distribution pattern from the SNr and GPe throughout the rostral and caudal extent of the PPN, we stereotaxically injected a channelrhodopsin virus with a fluorescent marker (AAV1-hSyn-ChR2-eYFP) into either the SNr or GPe of ChAT-Cre/Ai9-tdTomato mice (*Figure 1A and C*). We sliced 200 μm sagittal slices from mouse brains three weeks after viral injection and cleared the brain slices using the CUBIC clearing method (*Susaki et al., 2015*). Using the red fluorescent ChAT-positive neurons to define the borders of the PPN, we found that SNr axons appear in both the rostral and caudal PPN. The SNr axons fill the rostral PPN evenly but appear more distinctly clustered in specific areas of the caudal PPN (*Figure 1B*), whereas GPe axonal projections more densely fill the caudal region and strikingly avoid the rostral region (*Figure 1D*). These anatomical findings predict that the SNr will inhibit both rostral and caudal PPN neurons, while the GPe will selectively inhibit the caudal PPN. However, this axon imaging method does not reveal whether these axons form functional synapses with the neurons in the PPN and does not provide information on the strength of inhibition from each structure onto specific PPN cell types. Therefore, to evaluate the functional connectivity and synaptic characteristics from these inhibitory basal ganglia output structures to regionally- and molecularly-defined PPN neural subpopulations, we used ex vivo whole-cell electrophysiology to systematically record the inhibitory input to the rostral and caudal PPN from the SNr and GPe in three

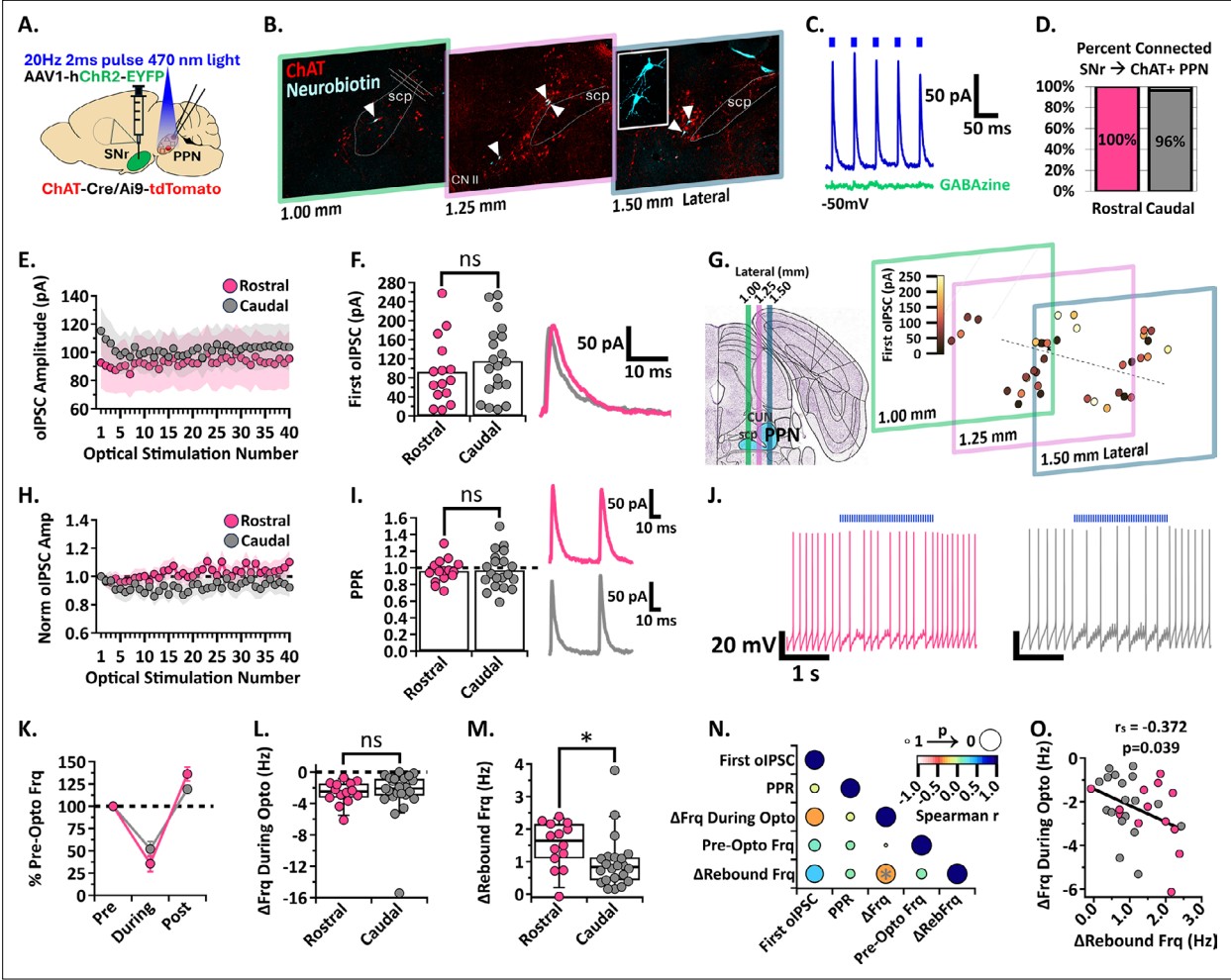

**Figure 2.** SNr inhibition of rostral and caudal ChAT +PPN neurons. (**A**) Experimental set up to identify red ChAT +PPN neurons for whole-cell patch clamp while stimulating ChR2-filled SNr axons [N=6]. (**B**) White arrowheads pointing to neurobiotin-filled patched neurons within the PPN across three 200 μm slices. (**C**) Example trace of the first five oIPSCs [blue] in the 2 second 20 Hz train inhibited by GABA-a receptor blocker, GABAzine [green], while holding the cell at –50 mV. (**D**) Percent connected among patched neurons in the rostral and caudal regions. (**E**) Average oIPSC amplitude at each of 40 optogenetic light pulses in n=15 rostral neurons and n=20 caudal neurons. (**F**) *Left*, Individual cell data for the first oIPSC amplitude and, *right*, example current traces. (**G**) Cell mapping of patched neuron locations with the first oIPSC amplitude represented by the color scale. (**H**) Normalized current amplitudes in E. (**I**) *Left*, Individual cell data for the PPR between the first two oIPSC amplitudes in the train and, *right*, example current traces. (**J**) Example voltage traces of action potential firing during a 2 s 20 Hz train stimulation in rostral (left) and caudal (right) neurons. (**K**) Percent of pre-optical stimulation firing frequency during stimulation and rebound in n=14 rostral vs n=23 caudal neurons. (**L**) Individual cell data for the absolute change in frequency during optical stimulation [ΔFrq During Opto]. (**M**) Individual cell data for the absolute change in rebound frequency post-stimulation [ΔRebFrq]; rostral vs. caudal p=0.0142. (**N**) Correlation analysis, color scale representing Spearman r [–1,1] and size representing p-value [1,0]. (**O**) Negative correlation between the absolute change in frequency during stimulation and post-stimulation rebound; $r=-0.372$, p=0.039. * $p<0.05$; bar graph data represent mean ± SEM; box plots show median line with boxes showing IQR and whiskers showing 9th and 91st percentiles.

genetic mouse lines (ChAT-cre, Vgat-cre, and Vglut2-cre) which label cholinergic, GABAergic, and glutamatergic PPN subpopulations, respectively (*Figure 1—figure supplement 1*).

## The SNr inhibits cholinergic neurons in both the rostral and caudal PPN to a similar extent but evokes stronger rebound in rostral neurons

To evaluate the functional connection between the SNr and cholinergic PPN neurons, we performed ex vivo whole-cell patch clamp electrophysiology paired with optogenetics to record synaptic currents in ChAT-Cre/Ai9-tdTomato mice injected with ChR2 in the SNr three weeks prior (*Figure 2A*). Neurons were targeted for whole-cell patch clamp in both the rostral and caudal PPN across three sagittal slices to evaluate the medial to lateral extent of the PPN (*Figure 2B*). To measure optically evoked

inhibitory post-synaptic currents (oIPSCs), cholinergic PPN neurons were held at –50 mV in whole-cell voltage clamp and blue (470 nm) light was applied (2ms pulse duration, 13 mW) to activate the ChR2-infected SNr axons. In connected neurons, inhibitory currents were recorded in the presence of glutamatergic receptor blockers (AP5, NBQX, CNQX) during stimulation (*Figure 2C*). In a subset of neurons, a GABA-A receptor blocker, GABAzine, was applied to confirm the oIPSCs were GABA-mediated (*Figure 2C*). GABAzine eliminated SNr-evoked inhibitory currents in most cells, but a small residual current remained in some cells (see *Methods* for details). We found that all rostral (n=19/19 cells, N=6 mice) and essentially all caudal (n=26/27, N=6) recorded cholinergic PPN neurons responded to SNr axon activation with an observable oIPSC (*Figure 2D*).

To determine whether the SNr input to the rostral and caudal PPN cholinergic neurons differed in synaptic strength, we measured the amplitude of each oIPSC in a 20 Hz train of 2ms optical stimulations applied for 2 s. There was no difference in amplitude of the first oIPSC in the stimulus train between rostral vs caudal cholinergic neurons (mean ± SEM; first oIPSC n=15 Rostral: 92.6±18 pA, n=20 Caudal: 115.2±17 pA; unpaired t-test, t=0.9041, df = 33, CI –28.4–73.7, p=0.3725; p=0.3725; *Figure 2E and F*). Going forward, the first oIPSC amplitude of the train will be referred to as the first oIPSC.

While the rostral and caudal anatomical separation of PPN neurons is commonly used, it is a relatively coarse division. Therefore, we carefully mapped the location of each recorded cholinergic PPN neuron by matching slices to their bregma reference in the Paxinos and Franklin's mouse brain atlas and aligning the midline of cholinergic neuron distribution which separates the loosely spread rostral neurons and densely packed caudal neurons (See *Methods* for details, *Figure 2G*). The strength of the SNr inhibitory connection is depicted by a color scale representative of the first oIPSC (*Figure 2G*, *right*). Throughout both the rostral and caudal PPN, cholinergic neurons receiving larger inputs from the SNr are intermixed with neurons receiving smaller inputs.

To compare short-term synaptic plasticity characteristics between rostral and caudal cholinergic PPN neurons, we normalized the amplitude of each current in the train to the first current. The oIPSC amplitude remained relatively constant with each subsequent pulse in both rostral and caudal cholinergic neurons (*Figure 2H*). In addition, the paired-pulse ratio (PPR) of the peak of the first two currents in the stimulation train did not differ (mean ± SEM; PPR n=15 Rostral: 0.96±0.04, n=20 Caudal: 0.97±0.05; unpaired t-test, t=0.1082, df = 33, CI –0.13–0.14, p=0.9145; *Figure 2I*). SNr synaptic inputs onto cholinergic neurons do not undergo significant short-term synaptic plasticity in either the rostral or caudal PPN.

Although oIPSC amplitude is indicative of the strength of a synapse, current size does not always correlate with functional impact on a neuron's action potential output (*Linders et al., 2022*; *Ambrosi and Lerner, 2022*; *Buchholz et al., 2023*). Because cholinergic PPN neurons display spontaneous firing (*Chen and Evans, 2024*; *Tubert et al., 2023*; *Takakusaki et al., 1997*; *Baksa et al., 2019*; *Kroeger et al., 2017*), we characterized the impact of SNr-mediated inhibition on action potential firing rate. While optically stimulating SNr axons with the same 20 Hz blue light protocol as above, we observed a decrease in firing rate in both rostral and caudal cholinergic PPN neurons (*Figure 2J–K*). However, there was no significant difference between the percent frequency change in rostral and caudal neurons during SNr axon stimulation (mean ± SEM; %Frequency n=14 Rostral: During Stimulation 35.62 ± 8.87%, Post Stimulation 136.1 ± 7.803%; n=23 Caudal: During Stimulation 52.11 ± 8.43%, Post Stimulation 119.1 ± 4.614%; two-way ANOVA, Stimulation$_{F(1, 35)=81.9}$ p<0.0001, Region$_{F(1, 35)=0.002052}$ p=0.9641, Interaction$_{F(1, 35)=3.273}$ p=0.0790; *Figure 2K*) or in the absolute decrease in firing frequency during optical stimulation [median (IQR); ΔFrq During Opto n=14 Rostral: –2.47 Hz (–3.37 to –1.44 Hz), n=23 Caudal: –2.10 Hz (–3.11 to –0.84 Hz); Mann Whitney, U=131, p=0.3602; *Figure 2L*]. These data show that the SNr equally inhibits action potential firing in rostral and caudal PPN cholinergic neurons.

Post-inhibitory rebound firing has been observed in the PPN as well as other brainstem structures (*Baksa et al., 2019*; *Granata and Kitai, 1991*; *Kang and Kitai, 1990*; *Evans et al., 2017*; *Villalobos and Basso, 2022*; *Rajaram et al., 2019*; *Zhu et al., 2022*; *Beaver and Evans, 2025*) and is associated with greater excitability, temporal encoding, and generation of oscillatory activity (*Sivaramakrishnan and Lynch, 2017*). Indeed, we found that the rostral and caudal cholinergic PPN neurons exhibited an increase in action potential firing after inhibition (*Figure 2K*). This is in agreement with previous literature showing rebound spikes following inhibitory postsynaptic

potentials recorded in PPN neurons of male rats during electrical stimulation of the SNr (*Granata and Kitai, 1991*; *Kang and Kitai, 1990*). However, rostral cholinergic neurons had greater rebound activity compared with caudal cholinergic neurons as measured by the absolute change in rebound frequency [median (IQR); ΔRebFrq n=14 Rostral: 1.64 Hz (1.00–2.17 Hz), n=22 Caudal: 0.83 Hz (0.42–1.14 Hz); Mann Whitney, U=79, p=0.0142; *Figure 2M*]. This aligns with previous findings showing that rostral cholinergic neurons tend to have stronger intrinsic rebound activity compared to caudal neurons (*Baksa et al., 2019*). Importantly, we show that such rebound can be evoked by synaptic inhibition from the SNr.

To detect trends among SNr connectivity characteristics and intrinsic electrophysiological properties of cholinergic PPN neurons, we performed a correlation analysis to determine the strength and directionality of the relationship among five parameters [first oIPSC, PPR, ΔFrq During Opto, pre-optical stimulation frequency (Pre-Opto Frq), and ΔRebFrq; *Figure 2N*]. We found a weak but significant negative correlation between the frequency decrease during SNr axon stimulation and the rebound frequency increase (Spearman *r*=−0.372, p=0.039, *Figure 2N and O*). This finding is expected since strong inhibition is more effective at producing rebound activity than weak inhibition. However, rostral PPN cholinergic neurons do not receive significantly stronger inhibition from the SNr compared with caudal cholinergic neurons (*Figure 2F and L*). Therefore, larger rebound frequency in rostral neurons is likely due to differences in intrinsic cellular properties rather than synaptic properties (*Baksa et al., 2019*). Put together, these findings show that the SNr inhibits rostral and caudal cholinergic PPN neurons to a similar extent but evokes stronger rebound firing in the rostral neurons.

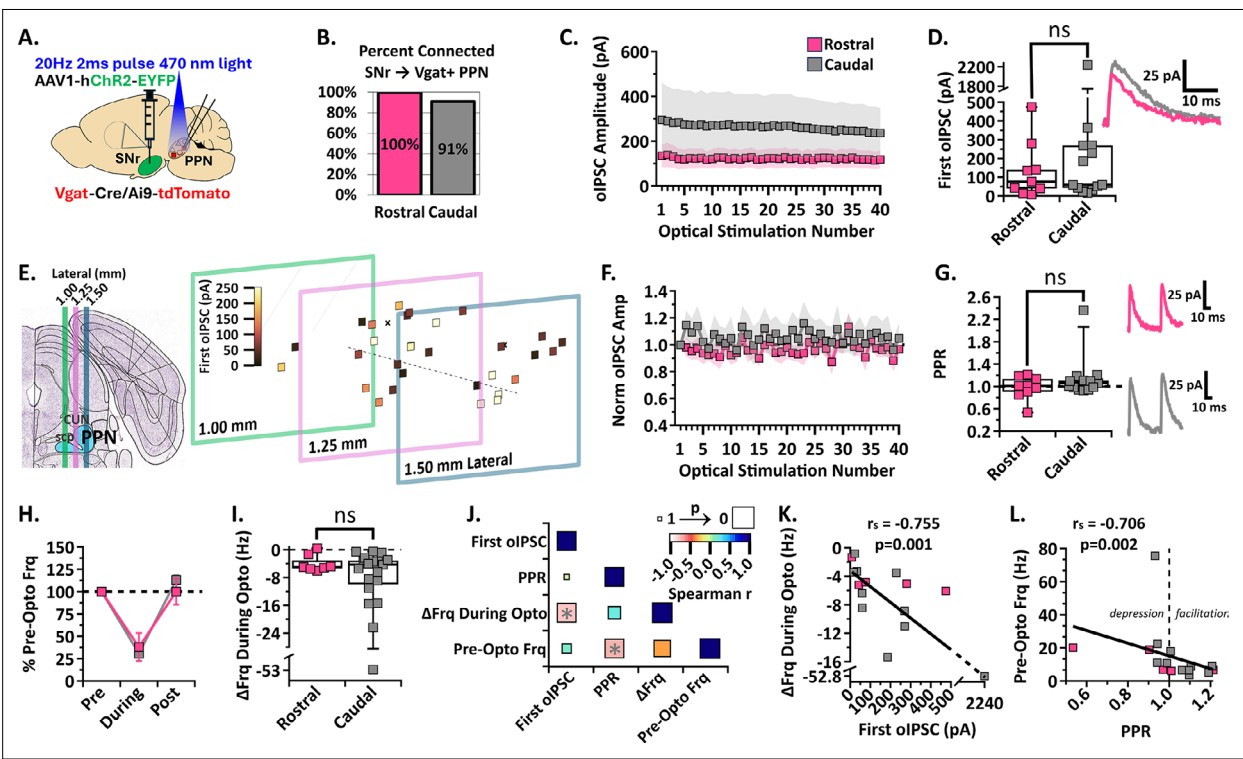

**Figure 3.** SNr inhibition of rostral and caudal Vgat +PPN neurons. (**A**) Experimental set up to identify red Vgat +PPN neurons for whole-cell patch clamp while stimulating ChR2-filled SNr axons [N=6]. (**B**) Percent connected among patched neurons in the rostral and caudal regions. (**C**) Average oIPSC amplitude at each of 40 optogenetic light pulses in n=9 rostral neurons and n=13 caudal neurons. (**D**) *Left*, Individual cell data for the first oIPSC amplitude and, *right*, example current traces. (**E**) Cell mapping of patched neuron locations with the first oIPSC amplitude represented by the color scale. (**F**) Normalized current amplitudes in C. (**G**) *Left*, Individual cell data for the PPR between the first two oIPSC amplitudes in the train and, *right*, example current traces. (**H**) Percent of pre-optical stimulation firing frequency [% Pre-Opto Frq] during stimulation and rebound in n=7 rostral and n=19 caudal neurons. (**I**) Individual cell data for the absolute change in frequency during optical stimulation [ΔFrq During Opto]. (**J**) Correlation analysis, color scale representing Spearman r [–1,1] and size representing p-value [1,0]. (**K**) Negative correlation between the absolute change in frequency during stimulation and first oIPSC amplitude; *r*=−0.755, p=0.001. (**L**) Negative correlation between the pre-optical stimulation firing frequency and the PPR; *r*=−0.706, p=0.002. Box plots show median line with boxes showing IQR and whiskers showing 9th and 91st percentiles.

## The SNr inhibits GABAergic neurons in both the rostral and caudal PPN to a similar extent

While there are several rabies tracing studies suggesting SNr axons project to the cholinergic and glutamatergic PPN neurons (*Zhao et al., 2022*; *Roseberry et al., 2016*; *Huerta-Ocampo et al., 2021*; *Dautan et al., 2021*; *Caggiano et al., 2018*), SNr projections to the GABAergic PPN neurons are inferred from a study of the entire mesencephalic locomotor region (*Roseberry et al., 2016*) and a characterization of their synaptic strength has not been conducted. Therefore, we evaluated the synaptic characteristics between the SNr and GABAergic neurons in the rostral and caudal PPN. In Vgat-Cre/Ai9-tdTomato mice, we injected ChR2 into the SNr to optically stimulate axons over the PPN while patching Vgat +GABAergic PPN neurons (*Figure 3A*). We post-hoc stained the slices for ChAT +neurons to verify the location of all patched GABAergic neurons within the cholinergic PPN. We found that all rostral (n=12/12, N=6) and most caudal (n=20/22, N=6) recorded GABAergic PPN neurons received inhibitory input from the SNr (*Figure 3B*).

Although there was no major difference in the proportion of GABAergic neurons receiving inhibitory input from the SNr in the rostral vs caudal PPN, we compared the current amplitudes to characterize the synaptic strength and pattern of this connection. The oIPSC amplitudes across the 2 s stimulation as well as the first oIPSC showed no difference between rostral and caudal GABAergic neurons [median (IQR); first oIPSC n=9 Rostral: 76.5 pA (27.0–209.9 pA), n=13 Caudal: 59.7 pA (37.7–269.7 pA); Mann Whitney, U=50, p=0.6005; *Figure 3C and D*]. We also did not observe significant differences in short-term synaptic plasticity between rostral and caudal GABAergic neurons [median (IQR); PPR n=9 Rostral: 1.01 (0.88–1.16), n=13 Caudal: 1.06 (0.96–1.15); Mann Whitney, U=48, p=0.5123; *Figure 3F and G*]. Defining short-term depression as a PPR <0.95 and short-term potentiation as PPR >1.05, we found 3/9 rostral and 7/13 caudal GABAergic PPN neurons showed short-term potentiation, while 3/9 rostral and 3/13 caudal GABAergic PPN neurons showed short-term depression. These data suggest additional heterogeneity within the GABAergic PPN subpopulation. Similar to the SNr connection with cholinergic PPN neurons, SNr input to the GABAergic PPN does not result in significant short-term synaptic plasticity and does not differ in amplitude or characteristics between rostral and caudal regions.

Because GABAergic PPN neurons display spontaneous firing (*Kroeger et al., 2017*), we recorded action potential firing in current clamp while optically stimulating the SNr axons. The percent inhibition was not significantly different between rostral and caudal neurons (mean ± SEM; %Frequency n=7 Rostral: During Stimulation 37.91 ± 15.91%, Post Stimulation 100.0 ± 14.57%; n=19 Caudal: During Stimulation 30.68 ± 8.03%, Post Stimulation 113.2 ± 5.187%; two-way ANOVA, Stimulation$_{F(1, 24)=44.63}$ p<0.0001, Region$_{F(1, 24)=0.09642}$ p=0.7588, Interaction$_{F(1, 24)=0.8841}$ p=0.3565; *Figure 3H*). There was also no difference in the absolute frequency change during optical stimulation between rostral and caudal neurons [median (IQR); ΔFrq During Opto n=7 Rostral: –5.03 Hz (–5.55 to –1.34 Hz), n=19 Caudal: –4.29 Hz (–11.02 to –2.94 Hz); Mann Whitney, U=55, p=0.5336; *Figure 3I*]. These data show that the SNr equally inhibits action potential firing in rostral and caudal GABAergic PPN neurons and does not evoke significant rebound firing in either neural population.

Overall, we found that the SNr inhibits the rostral and caudal GABAergic PPN neurons to a similar extent, but observed greater variability in synaptic strength and impact on neuronal action potential firing among caudal GABAergic neurons. One caudal subgroup responded with very small (<60 pA) amplitude currents and another subgroup responded with currents more than triple that amplitude, with one neuron responding with a 2.2 nA inhibitory current. We mapped the location of each recorded GABAergic neuron, as described above, and found no specific location corresponding with the neurons receiving stronger SNr inhibition across the PPN landscape (*Figure 3E*). This strong variability suggests that GABAergic PPN neurons may be divided into functionally distinct neural types that do not correspond to their rostrocaudal anatomical location. To determine if the neurons receiving larger inhibitory currents have distinct characteristics, we performed a correlation analysis (*Figure 3J*). As expected, we found that the absolute frequency reduction during inhibition is correlated with the first oIPSC amplitude recorded in a cell (Spearman r=−0.755, p=0.001; *Figure 3K*). Surprisingly, however, we found a significant negative correlation between PPR and the pre-optical stimulation frequency (Pre-Opto Frq, *Figure 3L*). Specifically, input to slower-firing neurons displayed greater short-term synaptic facilitation. This finding supports the idea that multiple functional neural subpopulations may be present within the GABAergic PPN population. Together, these results show that the SNr

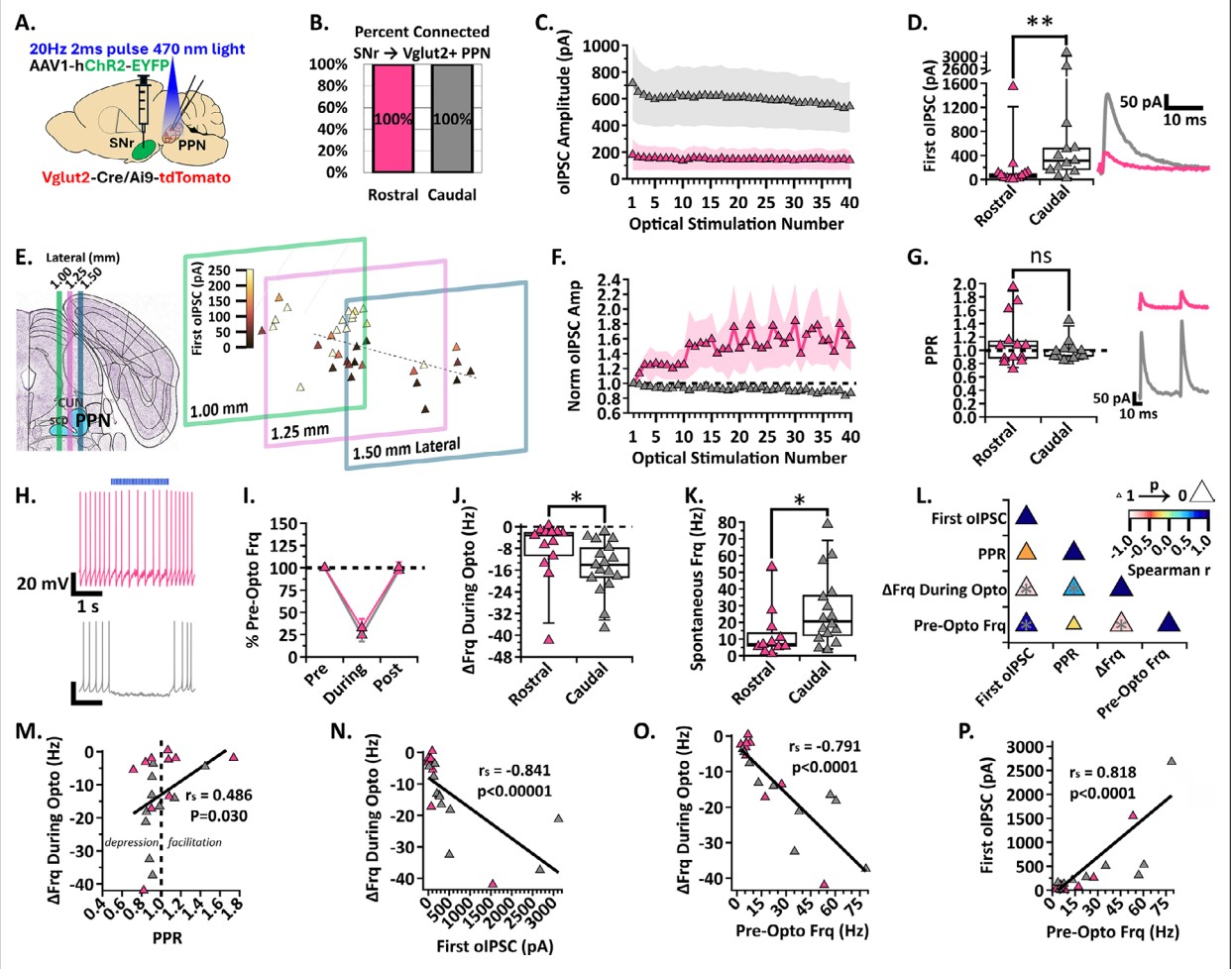

**Figure 4.** SNr inhibition of rostral and caudal Vglut2 +PPN neurons. (**A**) Experimental setup to identify red Vglut2 +PPN neurons for whole-cell patch clamp while stimulating ChR2-filled SNr axons [N=6]. (**B**) Percent connected among patched neurons in the rostral and caudal regions. (**C**) Average oIPSC amplitude at each of 40 optogenetic light pulses in n=13 rostral neurons and n=13 caudal neurons. (**D**) *Left*, Individual cell data for the first oIPSC amplitude and, *right*, example current traces; p=0.0035. (**E**) Cell mapping of patched locations with the first oIPSC amplitude represented by the color scale. (**F**) Normalized current amplitudes in C. (**G**) *Left*, Individual cell data for the PPR between the first two oIPSC amplitudes in the train and, *right*, example current traces. (**H**) Example voltage traces of action potential firing during a 2 s 20 Hz train stimulation in rostral and caudal neurons, top to bottom. (**I**) Percent of pre-optical stimulation firing frequency [% Pre-Opto Frq] during stimulation and rebound in n=13 rostral and n=17 caudal neurons. (**J**) Individual cell data for the absolute change in frequency during optical stimulation [ΔFrq During Opto]; p=0.0197. (**K**) Spontaneous frequency in n=11 rostral and n=16 caudal neurons; p=0.0343. (**L**) Correlation analysis, color scale representing Spearman r [–1,1] and size representing p-value [1,0]. (**M**) Positive correlation between the absolute change in frequency during stimulation and PPR, r=0.486, p=0.030. (**N**) Negative correlation between the absolute change in frequency during stimulation and first oIPSC amplitude; r=−0.841, p<0.00001. (**O**) Negative correlation between the absolute change in frequency during stimulation and pre-optical stimulation frequency; r=−0.791, p<0.0001. (**P**) Positive correlation between the first oIPSC amplitude and pre-optical stimulation frequency; r=0.818, p<0.0001. * p<0.05, ** p<0.01; box plots show median line with boxes showing IQR and whiskers showing 9th and 91st percentiles.

functionally inhibits GABAergic neurons across the rostrocaudal extent of the PPN, but suggest the presence of additional heterogeneity within the GABAergic PPN population.

## The SNr differentially inhibits the rostral and caudal glutamatergic PPN neurons

Previous rabies tracing anatomical studies have identified SNr inputs to glutamatergic PPN neurons (*Roseberry et al., 2016*; *Dautan et al., 2021*; *Caggiano et al., 2018*), but a full characterization of their synaptic strength and rostral and caudal connectivity has not been conducted. In Vglut2-Cre/Ai9-tdTomato mice, we injected ChR2 into the SNr to optically stimulate axons over the PPN while

patching Vglut2 +glutamatergic neurons in the PPN (*Figure 4A*). While holding cells at –50 mV and applying 470 nm blue light stimulation, neurons were identified as connected if oIPSCs were observed. Similar to our findings in cholinergic and GABAergic neurons, we found that all rostral (n=19/19, N=6) and all caudal (n=28/28, N=6) recorded glutamatergic neurons received inhibitory input from the SNr (*Figure 4B*).

Comparing the oIPSC amplitudes across the 20 Hz train, we discovered that the caudal glutamatergic neurons receive larger inhibitory currents than rostral glutamatergic neurons (*Figure 4C*). The median first oIPSC measured in caudal glutamatergic neurons was also significantly larger than those recorded in rostral neurons [median (IQR); first oIPSC n=13 Rostral: 61.6 pA (19.2–113.9 pA), n=13 Caudal: 313.5 pA (143.8–729.1 pA); Mann Whitney, U=29, p=0.0035; *Figure 4D*]. After post-hoc staining for the cholinergic PPN and mapping the location of each recorded glutamatergic PPN neuron as described above, we identified a 'hotspot' of strong SNr inhibition in a medial-caudal group of glutamatergic neurons (*Figure 4E*). This finding shows that a subset of caudal PPN neurons preferentially receives exceptionally strong SNr input.

To determine whether the SNr input to the rostral and caudal PPN shows differential short-term synaptic plasticity, we compared the normalized oIPSC amplitudes. We found that inhibitory inputs to the rostral glutamatergic neurons display short-term synaptic facilitation with amplitudes increasing in subsequent stimulations (*Figure 4F*). However, strong facilitation occurred in only a few neurons that drove the mean upward, and there was no significant difference between the PPR observed in rostral and caudal glutamatergic neurons when activating SNr axons [median (IQR); PPR n=13 Rostral: 1.07 (0.86–1.38), n=13 Caudal: 0.92 (0.89–1.02); Mann Whitney, U=77, p=0.7241; *Figure 4G*].

Because glutamatergic PPN neurons spontaneously fire (*Dautan et al., 2021*; *Kroeger et al., 2017*), we wanted to determine if larger inhibitory currents mediated larger decreases in the firing rate of caudal PPN neurons. Therefore, we recorded tonic action potential firing in glutamatergic PPN neurons while optically stimulating SNr axons (*Figure 4H*). Although inhibition measured as the percent of pre-optical stimulation firing frequency was not significantly different between rostral and caudal neurons (mean ± SEM; %Frequency n=13 Rostral: During Stimulation 32.62 ± 10.21%, Post Stimulation 100.3 ± 6.113%; n=17 Caudal: During Stimulation 24.71 ± 7.30%, Post Stimulation 97.48 ± 2.228%; two-way ANOVA, Stimulation$_{F(1, 28)=133.6}$ p<0.0001, Region$_{F(1, 28)=0.5214}$ p=0.4762, Interaction$_{F(1, 28)=0.1765}$ p=0.6776; *Figure 4I*), we found that the absolute frequency decrease was significantly larger in caudal glutamatergic neurons compared to rostral neurons during SNr stimulation [median (IQR); ΔFrq During Opto n=13 Rostral: –3.21 Hz (–12.21 to –1.95 Hz), n=17 Caudal: –14.08 Hz (–20.04 to –6.07 Hz); Mann Whitney, U=55, p=0.0197; *Figure 4J*]. To examine the discrepancy between the percent decrease and the absolute decrease in firing frequency, we compared the spontaneous firing frequency between rostral and caudal glutamatergic neurons in cells that had no holding current applied. We found that caudal glutamatergic neurons fire faster than rostral glutamatergic neurons [median (IQR); Spontaneous Frequency n=11 Rostral: 6.8 Hz (5.4–17.2 Hz), n=16 Caudal: 20.6 Hz (11.2–37.2 Hz); Mann Whitney, U=45, p=0.0343; *Figure 4K*]. Together, these findings show that fast-firing caudal glutamatergic PPN neurons receive larger SNr-mediated IPSCs and display greater decreases in their action potential firing.

Since we found that the caudal glutamatergic neurons receive larger inhibitory currents and display greater decreases in firing rate during SNr axon stimulation, we performed a correlation analysis to evaluate the strength of the relationship among the synaptic and firing characteristics (*Figure 4L*). Interestingly, there was a weak but significant positive correlation between the PPR and frequency change during stimulation, showing that input displaying short-term synaptic depression occurs in neurons with greater decreases in firing frequency (Spearman $r$=0.486, p=0.030; *Figure 4M*). As expected, SNr-mediated oIPSCs with larger amplitudes are correlated with greater decreases in firing frequency during stimulation (Spearman $r$=–0.841, p<0.00001; *Figure 4N*). Greater pre-optical stimulation firing frequency is correlated with larger frequency decreases during stimulation (Spearman $r$=–0.791, p<0.0001; *Figure 4O*). This finding is expected because the fast-firing neurons can be more strongly inhibited in terms of absolute decrease in action potential frequency (i.e. a floor effect). Surprisingly, however, pre-optical stimulation firing frequency is correlated with the first oIPSC amplitude (Spearman $r$=0.818, p<0.0001; *Figure 4P*), indicating that the SNr is more strongly connected to glutamatergic PPN neurons with faster firing rates. These correlations highlight strong selectivity in SNr projections to PPN glutamatergic neurons. Overall, these findings support that the SNr

selectively targets and strongly inhibits fast-firing caudal glutamatergic PPN neurons compared to rostral neurons, and that there is a medial-caudal hotspot where SNr inhibition of PPN glutamatergic neurons is particularly strong.

## The SNr most strongly inhibits caudal glutamatergic PPN neurons

Most of the previous PPN circuit work has been done with rabies tracing. Aside from technical limitations, rabies tracing can only indicate the presence of synaptic connections. Our electrophysiological findings comprehensively characterize the strength and characteristics of the SNr input to each of the PPN cell types in both the rostral and caudal regions. Using this comprehensive dataset, we compared SNr input across the different PPN cell types in each region under the same electrophysiological conditions. We found that the median amplitude of the first SNr-mediated oIPSC was the same among rostral PPN neurons irrespective of cell type [median (IQR); first oIPSC n=15 ChAT+: 74.7 pA (44.5–136.5 pA), n=9 Vgat+: 76.5 pA (27.0–209.9 pA), n=13 Vglut2+: 61.6 pA (19.2–113.9 pA); Kruskal-Wallis, $H_{(2,34)}$=0.7002 p=0.7046; *Figure 5A*]. The impact of these inhibitory inputs on neuronal firing did not differ with cell type as shown by the absolute change in frequency during SNr stimulation [median (IQR); ΔFrq During Opto n=14 ChAT+: –2.47 Hz (–3.37 to –1.44 Hz), n=7 Vgat+: –5.03 Hz (–5.54 to –1.34 Hz), n=13 Vglut2+: –3.21 Hz (–12.21 to –1.95 Hz); Kruskal-Wallis, $H_{(2,31)}$=2.389 p=0.3028; *Figure 5B*].

We found that the SNr-mediated oIPSCs in caudal glutamatergic neurons were significantly larger than in cholinergic and GABAergic caudal PPN neurons [median (IQR); first oIPSC n=20 ChAT+: 109.3 pA (58.5–168.6 pA), n=12 Vgat+: 59.7 pA (37.7–269.7 pA), n=13 Vglut2+: 313.5 pA (143.8–729.1 pA); Kruskal-Wallis, $H_{(2,43)}$=8.717 p=0.0128; Dunn's test, ChAT +vs Vgat +p = 0.6409, ChAT +vs Vglut2 +p = 0.0042, Vgat +vs Vglut2 +p = 0.0295; *Figure 5C*]. Likewise, the impact of SNr inhibition on neuron firing rate was strongest in caudal glutamatergic neurons. The absolute frequency decrease during stimulation in caudal glutamatergic neurons was larger than both cholinergic and GABAergic neurons [median (IQR); ΔFrq During Opto n=23 ChAT+: –2.10 Hz (–3.11 to –0.84 Hz), n=19 Vgat+: –4.29 Hz (–11.02 to –2.94 Hz), n=17 Vglut2+: –14.08 Hz (–20.04 to –6.07 Hz); Kruskal-Wallis, $H_{(2,56)}$=24.99 p<0.0001; Dunn's test, ChAT +vs Vgat +p = 0.0049, ChAT +vs Vglut2 +p < 0.0001, Vgat +vs Vglut2 +p = 0.0332; *Figure 5D*]. While we found no significant difference between SNr-mediated oIPSCs recorded in GABAergic and cholinergic neurons (*Figure 5C*), the GABAergic neuron firing frequency was more strongly inhibited than cholinergic (*Figure 5D*). The greater reduction in firing frequency may be driven by the caudal GABAergic subgroup receiving particularly large oIPSCs (*Figure 3D*) and could be explained if GABAergic PPN neurons have higher input resistance than cholinergic neurons, as has been previously suggested (*Bordas et al., 2015*). Altogether, our findings show that the SNr most strongly inhibits the glutamatergic neurons, followed by GABAergic neurons in the caudal PPN, and more weakly inhibits rostral PPN neurons without any cell type bias (*Figure 5E*).

## The GPe preferentially, but weakly, inhibits a subset of caudal GABAergic and glutamatergic PPN neurons

GPe projections to the cholinergic and glutamatergic PPN neurons have been shown in previous rabies tracing studies (*Roseberry et al., 2016*; *Huerta-Ocampo et al., 2021*; *Dautan et al., 2021*; *Caggiano et al., 2018*), but projections to the GABAergic MLR neurons were not detected (*Roseberry et al., 2016*). To evaluate the synaptic strength of these connections and determine regional connectivity, we repeated the previous whole-cell patch clamp experiments paired with optogenetics, but this time injecting ChR2 into the GPe of ChAT-Cre/Ai9-tdTomato, Vgat-Cre/Ai9-tdTomato, and Vglut2-Cre/tdTomato mice (*Figure 6A*). If oIPSCs were observed in response to optical stimulation, the neuron was identified as connected. In a subset of neurons, a GABA-A receptor blocker, GABA-zine, was applied to confirm the oIPSCs were GABA-mediated (see *Methods* for details; *Figure 6B*). We recorded GPe-mediated oIPSCs in all three cell types of the PPN. As predicted by the low density of GPe axons present in the rostral PPN (*Figure 1D*), very few rostral PPN neurons received synaptic input from the GPe. This was true within each molecularly-defined PPN cell type [caudal vs rostral % connected; ChAT+: 30% vs 0% (n=9/30 vs 0/7), Vgat+: 68% vs 38% (n=21/31 vs 6/16), Vglut2+: 69% vs 26% (n=24/35 vs 5/19); *Figure 6Ci–iii*]. These findings show that the GPe preferentially targets caudal PPN neurons compared to rostral neurons. Since so few rostral PPN neurons received GPe inhibition, we focused our characterization of GPe inhibition to the caudal PPN.

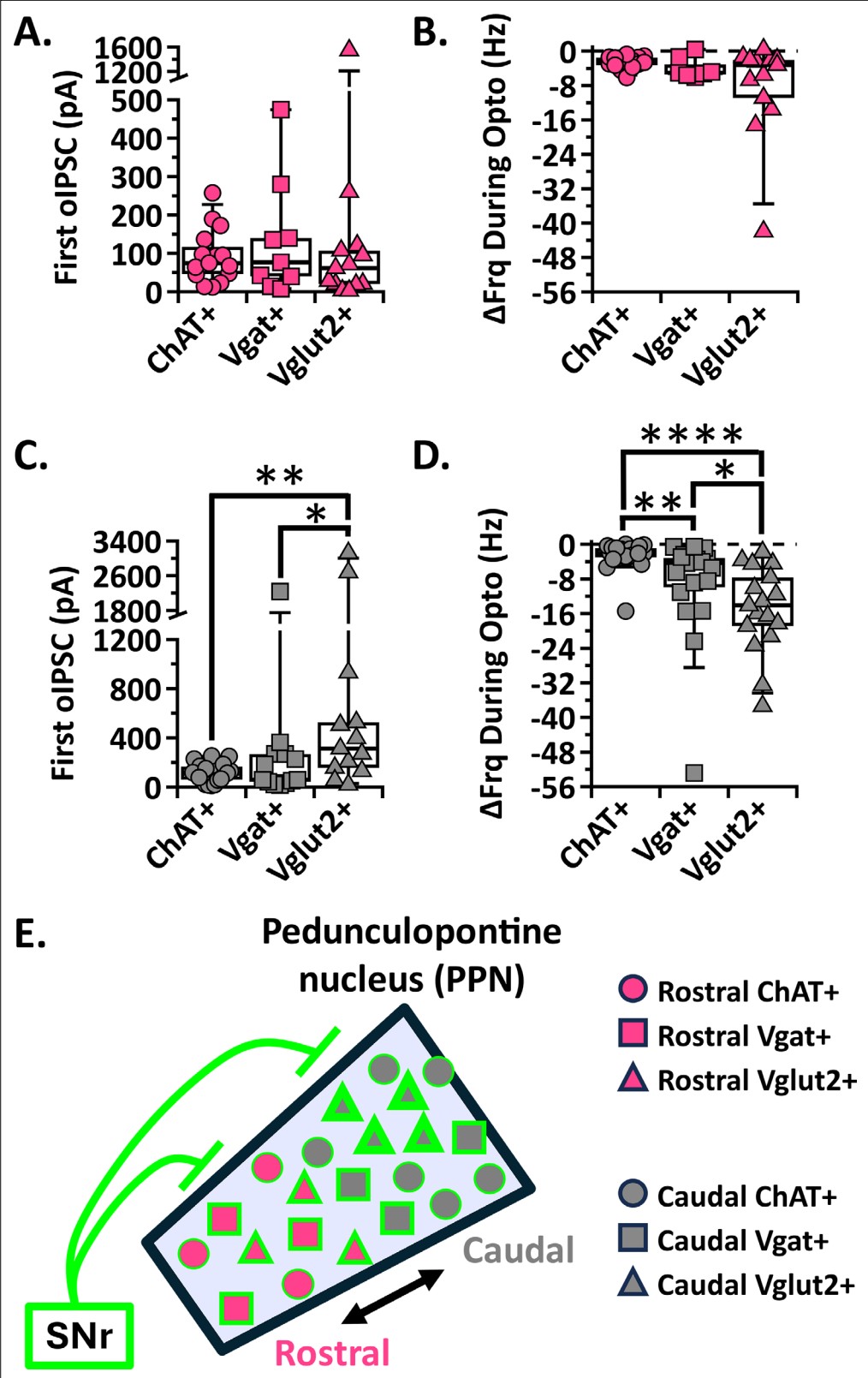

**Figure 5.** The SNr most strongly inhibits caudal glutamatergic PPN neurons. (**A, C**) Individual cell data for the first oIPSC amplitude recorded in each cell type for rostral and caudal PPN neurons, respectively. (**B, D**) Individual cell data for the absolute change in frequency during stimulation in each cell type for rostral and caudal PPN neurons, respectively. (**E**) Graphical depiction of SNr stimulation results. * p<0.05, ** p<0.01, **** p<0.0001; box plots show median line with boxes showing IQR and whiskers showing 9th and 91st percentiles.

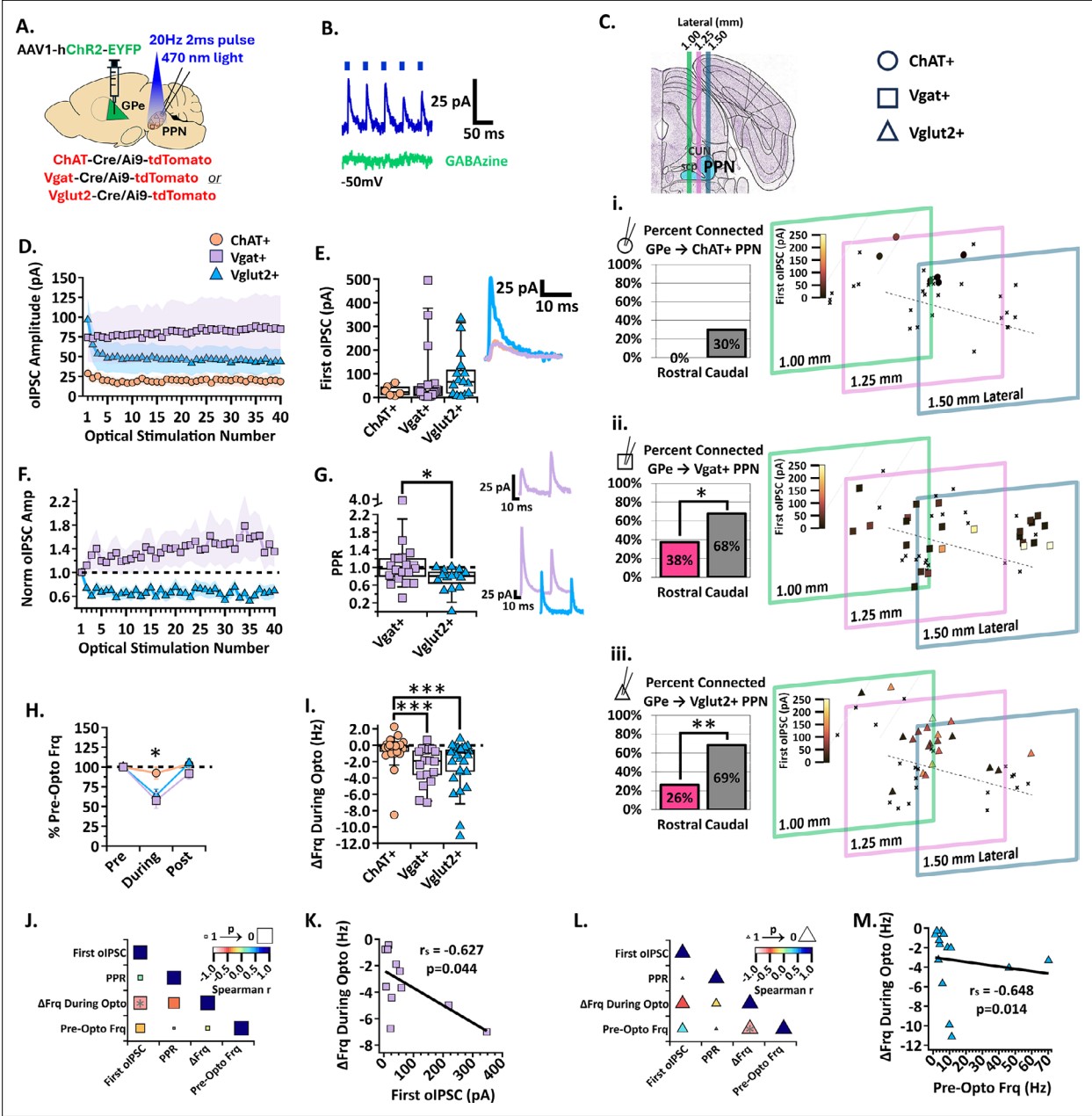

**Figure 6.** GPe inhibition of the three PPN cell types. (**A**) Experimental set up to identify red ChAT+, Vgat+, and Vglut2 +PPN neurons for whole-cell patch clamp while stimulating ChR2-filled GPe axons [N=6]. (**B**) Example trace of the first five oIPSCs in the 2 s 20 Hz train [blue] inhibited by GABA-a receptor blocker, GABAzine [green], while holding the cell at −50 mV. (**C**) *Left*, Percent connected among patched neurons in the rostral and caudal regions and, *right*, cell mapping of patched locations with the first oIPSC amplitude represented by the color scale. *Top to bottom*, i. ChAT+, ii. Vgat+, and iii. Vglut2 +datasets. (**D**) Average oIPSC amplitude at each of 40 optogenetic light pulses in n=6 ChAT+, n=19 Vgat+, and n=15 Vglut2 +caudal PPN neurons. (**E**) *Left*, Individual cell data for the first oIPSC amplitude and, *right*, example current traces. (**F**) Normalized current amplitudes in C. (**G**) *Left*, Individual cell data for the PPR between the first two oIPSC amplitudes in the train; p=0.0206. R*ight, top*, example current trace of short-term synaptic facilitation in VgAT +neurons. *Right, bottom*, example current traces of short-term synaptic depression in Vgat +and Vglut2 +neurons. (**H**) Percent of pre-optical stimulation firing frequency [%Pre-Opto Frq] during and post-stimulation in n=25 ChAT+, n=18 Vgat+, and n=29 Vglut2 +caudal PPN neurons. (**I**) Individual cell data for the absolute change in frequency during stimulation [ΔFrq During Opto]. (**J**) Correlation analysis for Vgat + neurons, color scale representing Spearman r [–1,1] and size representing p-value [1,0]. (**K**) Negative correlation between the absolute change in frequency during stimulation and first oIPSC amplitude; r=−0.627, p=0.044. (**L**) Correlation analysis for Vglut2 +neurons. (**M**) Negative correlation between the absolute change in frequency during stimulation and the pre-stimulation firing frequency; r=−0.648, p=0.014. *p<0.05, **p<0.01; box plots show median line with boxes showing IQR and whiskers showing 9th and 91st percentiles.

The percentage of connected caudal neurons was greater than the percentage of connected rostral neurons in both GABAergic and glutamatergic neurons (one-sided Fisher's exact test; Vgat: OR = 0.2857, CI = 0.09190–0.9391, p=0.047; Vglut2: OR = 0.1637, CI = 0.05069–0.5767, p=0.0033; *Figure 6Cii and iii*), but we found no statistical difference between the rostral and caudal connectivity in cholinergic neurons (OR = 0, CI = 0–1.301, p=0.1150; *Figure 6Ci*). The number of connected caudal cholinergic neurons was not significantly greater than its 0% rostral connectivity (*Figure 6Ci*). In addition, the average oIPSC amplitude across the train and the first oIPSC measured in the connected cholinergic neurons was small [median (IQR); first oIPSC n=6 ChAT+: 22.6 pA (9.75–50.20 pA); *Figure 6D and E*]. Among recorded caudal cholinergic PPN neurons, both connected and non-connected, these small inhibitory currents displayed negligible effects on the neuronal firing frequency of the cholinergic PPN population in both the percent of pre-optical stimulation frequency and absolute frequency change [median (IQR); n=29 ChAT+: %Frequency During Stimulation 96.67% (85.1 to 100.7%); ΔFrq During Opto –0.16 Hz (–0.90–0.03 Hz); *Figure 6H and I*]. These findings indicate that the GPe has essentially no direct effect on the cholinergic PPN neural population. For this reason, we exclude GPe inhibition of cholinergic PPN neurons in the synaptic characterizations shown in *Figure 6F and H*. Together, we show that the GPe preferentially inhibits caudal GABAergic and glutamatergic neurons while largely avoiding rostral and cholinergic PPN neurons.

To determine whether the synaptic strength and characteristics from the GPe to the PPN differed between GABAergic and glutamatergic PPN subtypes, we evaluated the amplitude of the GPe-mediated oIPSCs during the 2 s 20 Hz optical stimulation. Both GABAergic and glutamatergic neurons responded to GPe stimulation with oIPSCs ranging from ones to hundreds of picoamperes (*Figure 6D*). While there was no significant difference in the first oIPSC measured in GABAergic and glutamatergic neurons [median (IQR); first oIPSC n=19 Vgat+: 24.77 pA (6.72–54.17 pA), n=15 Vglut2+: 65.71 pA (14.50–133.4 pA); Kruskal-Wallis, $H_{(2,37)}$=3.050, p=0.2176; *Figure 6E*], the normalized amplitudes show that GPe input to GABAergic neurons facilitates with subsequent stimulations, while the input to glutamatergic neurons depresses (*Figure 6F*). The GPe-mediated inhibitory input to glutamatergic neurons displayed a significantly lower PPR than to GABAergic neurons [median (IQR); PPR n=19 Vgat+: 0.97 (0.76–1.2), n=15 Vglut2+: 0.81 (0.55–0.93); Mann-Whitney, U=76, p=0.0206; *Figure 6G*]. While short-term synaptic plasticity patterns were variable among GABAergic neurons, with some depressing and others facilitating, GPe input to glutamatergic neurons generally depresses.

Short-term synaptic depression can dampen the impact that repeated IPSCs have on action potential output. Therefore, we compared the firing frequency change in each cell type during optogenetic stimulation of GPe axons. The PPN GABAergic neuron population firing rate was decreased to about 58% of their pre-stimulation firing with a median frequency decrease of 1.92 Hz during stimulation (*Figure 6H and I*). Similarly, the glutamatergic neuron population firing rate decreased to about 64% of their pre-stimulation firing with a median frequency decrease of 0.89 Hz during stimulation (*Figure 6H and I*). We found no statistical difference between GPe inhibition of firing frequency in caudal GABAergic and glutamatergic PPN neurons (mean ± SEM; %Frequency n=29 ChAT+: During Stimulation 92.39 ± 7.59%, Post Stimulation 104.10 ± 5.23%; n=18 Vgat+: During Stimulation 57.70 ± 10.11%, Post Stimulation 91.81 ± 6.98%; n=24 Vglut2+: During Stimulation 64.15 ± 7.63%, Post Stimulation 104.7 ± 5.39%; two-way ANOVA, Stimulation$_{F(1, 69)=25.46}$ p<0.0001, Cell type$_{F(1, 69)=4.444}$ p=0.0153, Interaction$_{F(1, 69)=2.805}$ p=0.0674; Fisher's LSD, df = 138, %Frequency During Stimulation: ChAT +vs. Vglut2 +p = 0.0035, ChAT +vs. Vgat +p = 0.0016, Vglut2 +vs. Vgat +p = 0.5378, %Frequency Post Stimulation: ChAT +vs. Vglut2 +p = 0.8883, ChAT +vs. Vgat +p = 0.2838, Vglut2 +vs. Vgat +p = 0.2184; median (IQR); ΔFrq During Opto n=25 ChAT+: –0.16 Hz (–0.90–0.03 Hz), n=18 Vgat+: –1.92 Hz (–3.85 to –0.60 Hz), n=29 Vglut2+: –0.89 Hz (–3.28 to –0.43 Hz); Kruskal-Wallis, $H_{(2,69)}$=17.21, p=0.0002; Dunn's test, ChAT +vs Vgat +p = 0.0002, ChAT +vs Vglut2 +p = 0.0009, Vgat vs Vglut2 +p = 0.4020; *Figure 6H&I*). These findings show that the GPe inhibits GABAergic and glutamatergic neural populations to a similar extent.

To determine if there are significant relationships among the synaptic and firing characteristics, we performed a correlation analysis. Among GABAergic neurons receiving GPe inhibition, we found that the amplitude of the first oIPSC is significantly correlated with the absolute change in frequency during stimulation (Spearman *r*=–0.627, p=0.044, *Figure 6K*). Although GPe input to GABAergic neurons exhibited various PPRs, the correlation analyses revealed no significant relationship between PPR and absolute change in frequency during stimulation (*Figure 6J*). These findings suggest that the

initial oIPSC amplitude of GPe input to GABAergic PPN neurons has a stronger relationship with firing rate than short-term synaptic plasticity does.

As nearly all GPe inputs to glutamatergic neurons exhibited short-term synaptic depression, we expected the correlation analysis to reveal a significant relationship between PPR and the change in firing frequency. However, similar to the GPe input to GABAergic neurons, PPR was not predictive of impact on neuronal activity (*Figure 6L*). GPe inhibition of the glutamatergic neurons showed a significant correlation between the absolute change in frequency during stimulation and pre-stimulation firing frequency (Spearman $r=-0.648$, p=0.014, *Figure 6M*). However, because the inhibitory synaptic current amplitude did not correlate with firing frequency, this correlation is likely due to a floor effect in which faster firing neurons have a greater capacity for reductions in firing frequency. Together, these findings show that the GPe preferentially inhibits caudal GABAergic and glutamatergic PPN neurons.

## In vivo activation of GPe and SNr axons in the PPN shows differential effects on locomotion and valence

Once we established that the SNr and GPe inhibit different profiles of regionally and molecularly defined subpopulations of the PPN using ex vivo optogenetic electrophysiology, we wanted to evaluate the behavioral consequences of selectively stimulating these inputs in vivo. Direct optogenetic activation of PPN subpopulations has been previously shown to either promote or inhibit motion, sometimes with contradictory results (*Roseberry et al., 2016*; *Dautan et al., 2021*; *Caggiano et al., 2018*; *Goñi-Erro et al., 2023*; *Masini and Kiehn, 2022*; *Gut et al., 2022*; *Huang et al., 2024*; *Josset et al., 2018*; *Tello et al., 2024*; *Xiao et al., 2016*). To determine whether differential inhibitory inputs onto the PPN influenced locomotor activity, we bilaterally injected wild-type mice with ChR2 in either the SNr or GPe and implanted an optical fiber over the PPN (*Figure 7A&B*). Control mice were injected with an eGFP virus and implanted. After three weeks, we placed each mouse in the open field to measure gross locomotor behavior with and without optical stimulation of the SNr or GPe axons. We tracked the distance traveled for each mouse at baseline and during bilateral stimulation of either the SNr or GPe by applying blue light (473 nm, 4–4.5 mW) at 20 Hz with 2ms pulses (4% duty cycle), for 1 min at a time (*Figure 7C*). We found that stimulation of these two basal ganglia inputs to the PPN resulted in differential motor behaviors. Specifically, stimulation of SNr axons in the PPN increased distance traveled, while stimulation of GPe axons in the PPN decreased distance traveled [mean ± SEM; N=9 (3M, 6 F) Control (Ctrl): 2.22±0.24 m, N=8 (5M, 3 F) SNr: 4.11±0.51 m, N=9 (4M, 5 F) GPe: 0.57±0.18 m; Welch ANOVA$_{W(4.00, 15.86)=14.81}$, p<0.0001; Dunnett's test, Ctrl$_{4.25mW}$ vs SNr$_{4.25mW}$ p=0.0263, Ctrl$_{4.25mW}$ vs GPe$_{4.25mW}$ p=0.0002; *Figure 7*, *Figure 7—figure supplement 1*, *Videos 1 and 2*].

The PPN has been implicated in reward processing and stimulation of the PPN can be reinforcing (*Yoo et al., 2017*; *Xiao et al., 2016*; *Dautan et al., 2016*; *Zhang et al., 2024*; *Liu et al., 2023*; *Aitta-aho et al., 2018*). Therefore, we evaluated the effect of GPe or SNr axon stimulation in the PPN on valence in the real-time place preference (RTPP) task. In a three-chamber apparatus, the mouse could freely move between chambers for ten minutes. Optical stimulation of the SNr or GPe axons in the PPN was applied when the mouse entered the stimulated chamber and remained on at 20 Hz (4% duty cycle) until the mouse exited that chamber (*Figure 7E*). To reduce the motor effects of optical stimulation, this experiment was conducted with unilateral stimulation only. Interestingly, we again found opposite effects when stimulating the two basal ganglia inputs to the PPN. Mice avoided the stimulated chamber when the SNr axons were stimulated, but preferred the stimulated chamber when the GPe axons were stimulated (mean ± SEM; N=16 (6M, 10 F) Ctrl: 40.76 ± 2.0%, N=9 (5M, 4 F) SNr: 23.99 ± 5.2%, N=10 (5M, 5 F) GPe: 76.11 ± 3.7%; Welch ANOVA$_{W(4.00,16.95)=26.73}$, p<0.0001; Dunnett's test, Ctrl$_{4.25mW}$ vs SNr$_{4.25mW}$ p=0.0437, Ctrl$_{4.25mW}$ vs GPe$_{4.25mW}$ p<0.0001, Ctrl$_{4.25mW}$ vs SNr$_{0.25mW}$ p=0.0695, Ctrl$_{4.25mW}$ vs GPe$_{0.25mW}$ p=0.0467; *Figure 7F*, see *Videos 3 and 4*). To further control for potential motion confounds, we recorded the location preference of the mice in the absence of optical stimulation the day after the stimulated RTPP assay (regardless of whether they received low or high power on day 1). The mice were placed in the central, neutral chamber and their location was tracked for 10 min. During the first minute of this assay, 8 out of 9 mice who had SNr optical stimulation on the first day continued avoiding the stimulated chamber while 9 out of 10 mice who had GPe optical stimulation continued to prefer the previously stimulated chamber compared to the median percent time spent in the stimulated chamber for mice in the control condition median (IQR); N=16 (6M, 10 F) Ctrl: 47% (14.79–69.24 %), N=9 (5 M, 4 F) SNr: 1.833% (0–41.75 %), N=10 (5M, 5 F)

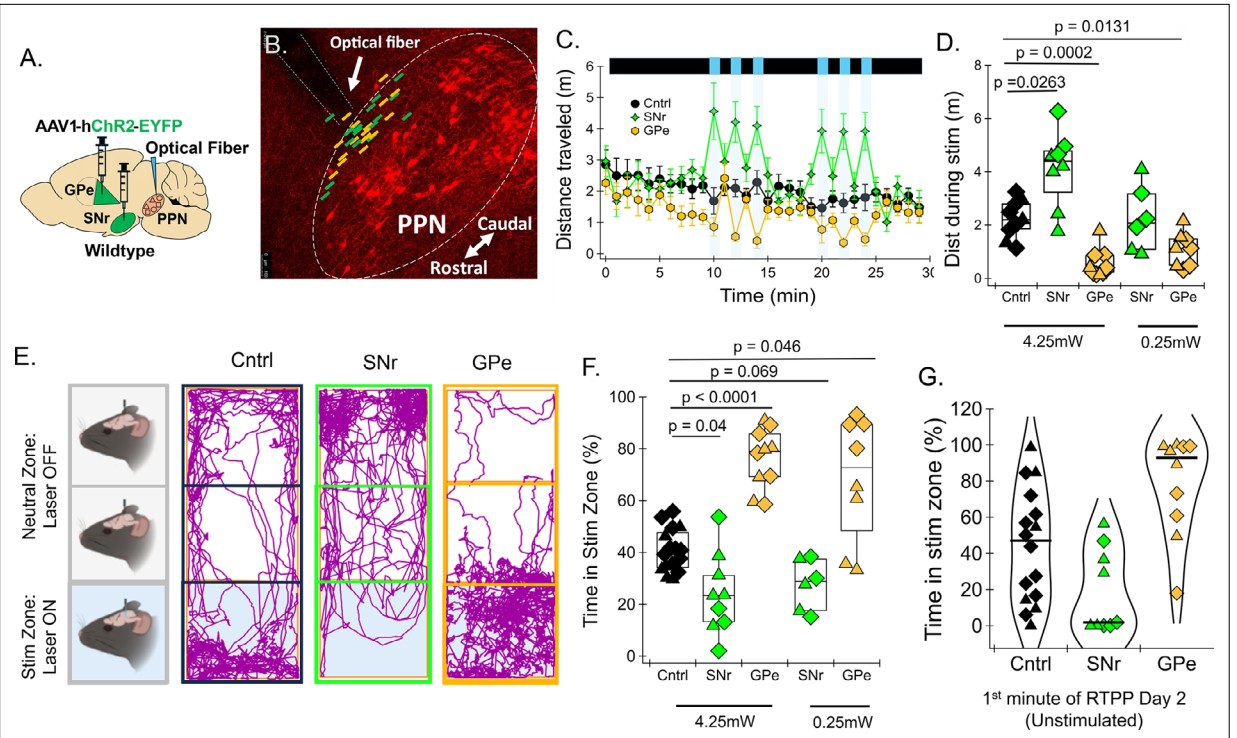

**Figure 7.** In vivo activation of GPe and SNr axons in the PPN shows differential effects on locomotion and opposite effects on valence. (**A**) Experimental set up to stimulate ChR2-filled SNr or GPe axons over the PPN in vivo. (**B**) Representative image of optical fiber tract overlaid with the approximate optical fiber placement for SNr- [green] and GPe- [orange] injected mice. (**C**) Distance traveled over time in an open field with 1 min 20 Hz 4.25 mW optical stimulations over the PPN in N=9 control (Ctrl) mice (black circles), N=8 mice injected with ChR2 in the SNr (green diamonds), and N=9 mice injected with ChR2 in the GPe (orange hexagons); vertical blue lines represent periods of optical stimulation. (**D**) Average distance traveled for each mouse across the six 1 min optical stimulations in the high (4.25 mW) and low laser power (0.25 mW) conditions. Marker shape represents male (triangle) and female (diamond) mice. (**E**) Representative mouse track tracings during real-time place preference task in a three-chamber box and continuously stimulating EGFP- or ChR2-filled axons over the PPN at 20 Hz in SNr- and GPe-injected mice when the mice are in the stimulation zone. (**F**) Percent time spent in the stimulation zone in N=16 control mice, N=9 mice injected with ChR2 in the SNr, and N=10 mice injected with ChR2 in the GPe with 4.25 or 0.25 mW laser power. (**G**) Percent time spent in the stimulation zone during the first minute reintroduced to the RTPP box on day 2 of RTPP with the laser off (no chamber is stimulated) Black line = median. * p<0.05, ** p<0.01, *** p<0.001, **** p<0.0001; box plots show median line with boxes showing IQR and whiskers showing 9th and 91st percentiles. See related *Videos 1–4*.

The online version of this article includes the following figure supplement(s) for figure 7:

**Figure supplement 1.** Speed measurements across age and sex.

**Figure supplement 2.** Opposite place preference despite similar motor responses to nigral axon stimulation in the PPN of DAT-Cre and wild-type mice.

**Figure supplement 3.** Rostral vs caudal implant site alters SNr axon stimulation effects.

**Figure supplement 4.** Graphical abstract summarizing key findings.

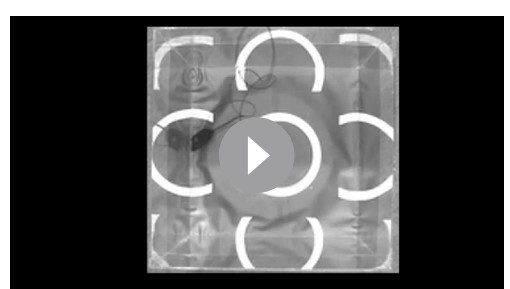

**Video 1.** Mouse behavior during high power constitutive substantia nigra axon stimulation in the PPN during open field.

https://elifesciences.org/articles/102308/figures#video1

GPe: 92.83% (58–99.21 %); Kruskal-Wallis$_{H=14.32}$, p=0.0008; Dunn's test, Ctrl vs SNr p=0.1433, Ctrl vs GPe 0.0312; *Figure 7G*.

In addition to sending projections to the PPN, the GPe and SNr send axon collaterals to multiple other brain areas (*McElvain et al., 2021*; *Mastro et al., 2014*; *Dong et al., 2021*; *Lilascharoen et al., 2021*). Therefore, it is possible that stimulating these axons in the PPN could cause antidromic stimulation of the GPe and SNr cell bodies, resulting in the inhibition of non-PPN brain regions. However, it has recently been shown that 0.25 mW laser power stimulation of axons can

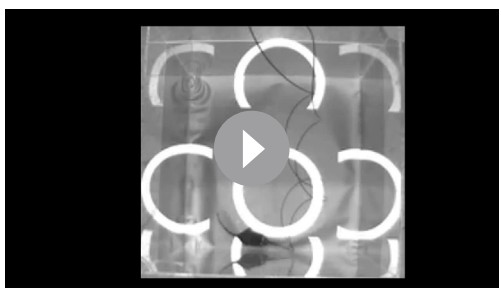

**Video 2.** Mouse behavior during high power GPe axon stimulation in the PPN during open field.
https://elifesciences.org/articles/102308/figures#video2

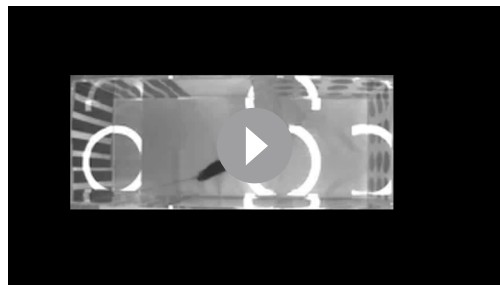

**Video 3.** Mouse behavior during high power constitutive substantia nigra axon stimulation in the PPN during real-time place preference task (striped side is stimulated side).
https://elifesciences.org/articles/102308/figures#video3

prevent antidromic activation of cell bodies (*Isett et al., 2023*). Therefore, we repeated these optogenetic behavioral experiments using 0.25 mW laser power. We found that the locomotion effect of SNr stimulation over the PPN was no longer present during low power stimulation, but GPe stimulation still significantly decreased locomotion (mean ± SEM; distance traveled N=9 (3M, 6 F) Ctrl: 2.22±0.24 m, N=6 (3M, 3 F) SNr: 2.24±0.50 m, N=8 (4M, 4 F) GPe: 1.09±0.22 m; Welch ANOVA$_{W(4.00, 15.86)=14.81}$; p<0.0001; Dunnett's test, Ctrl$_{4.25mW}$ vs SNr$_{0.25mW}$ p>0.999, Ctrl$_{4.25mW}$ vs GPe$_{0.25mW}$ p=0.0131; *Figure 7D*). In the RTPP task, SNr axon stimulation continued to evoke place aversion at the trend level and GPe axon stimulation continued to significantly evoke place preference (mean ± SEM; % time in stimulation zone N=16 (6M, 10 F) Ctrl: 40.76 ± 2.1%, N=6 (3M, 3 F) SNr: 27.72 ± 4.0%, N=8 (4M, 4 F) GPe: 68.41 ± 8.5%; Welch ANOVA$_{W(4.00,16.95)=26.73}$, p<0.0001; Dunnett's test, Ctrl$_{4.25mW}$ vs SNr$_{0.25mW}$ p=0.0695, Ctrl$_{4.25mW}$ vs GPe$_{0.25mW}$ p=0.0467; *Figure 7F*). These findings show that decreased locomotion and place preference are mediated by GPe inhibition of the PPN, while aversion is mediated by SNr inhibition of the PPN.

The increase in locomotion observed when stimulating the SNr axons in the PPN with high laser power is unexpected given that previous studies directly stimulating SNr neurons report decreases in movement (*Rizzi and Tan, 2019*; *Galaj et al., 2020*). Because injecting constitutive AAVs into the SNr can infect both GABAergic neurons and adjacent dopaminergic neurons of the substantia nigra pars compacta (SNc), we wanted to determine whether stimulating dopaminergic axons in the PPN enhanced locomotion and evoked aversion. To test this, we injected a cre-dependent ChR2 into the substantia nigra of DAT-cre mice (3 M heterozygotes and 3 F homozygotes) and ran open field and RTPP behavioral assays. We found that selective stimulation of dopaminergic axons in the PPN increased locomotion at high laser power (*Figure 7—figure supplement 2A*) and had no effect on locomotion at low power (*Figure 7—figure supplement 2C*). This finding suggests that dopaminergic neurons in the SNc send collateral projections to both the PPN and other locomotor-related structures (e.g. the striatum) such that stimulation of PPN-projecting SNc neurons increases locomotion. The effect of dopaminergic axon stimulation did not differ from whole substantia nigra stimulation in either high (mean ± SEM; N=8 (5M, 3 F) SNr$_{4.25mW}$: 4.107±0.51 m, N=6 (3M, 3 F) DAT-Cre$_{4.25mW}$: 3.778±0.28 m; unpaired t-test, t=0.5159, df = 12, CI = −1.715–1.059, p=0.6153; *Figure 7—figure supplement 2B*) or low laser power (mean ± SEM; N=6 (3M, 3 F) SNr$_{0.25mW}$: 2.238±0.58 m, N=9 (3M, 3 F) DAT-Cre$_{0.25mW}$: 1.984±0.30 m; unpaired t-test, t=0.4350, df = 10, CI = −1.556–1.047, p=0.6728; *Figure 7—figure supplement 2C*). On the other hand, we found that stimulation of dopaminergic axons in the PPN did not cause place aversion, but rather induced real-time place preference at both high and low laser power (*Figure 7—figure supplement 2D–F*). Selectively stimulating dopaminergic axons induced place preference behavior

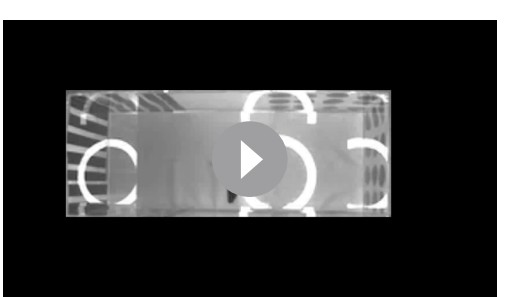

**Video 4.** Mouse behavior during high power GPe axon stimulation in the PPN during real-time place preference task (striped side is stimulated side).
https://elifesciences.org/articles/102308/figures#video4

significantly different from and opposite to whole substantia nigra stimulation at high (mean ± SEM; N=9 (5M, 4 F) $SNr_{4.25mW}$: 23.99 ± 5.18%, N=6 (3M, S3F) $DAT-Cre_{4.25mW}$: 70.83 ± 10.06%; unpaired t-test, t=4.545, df = 13, CI = 24.58–69.11, p=0.0005; *Figure 7—figure supplement 2E*) and low laser power (mean ± SEM; N=6 (3M, 3 F) $SNr_{0.25mW}$: 27.72 ± 4.0%, N=6 (3M, 3 F) $DAT-Cre_{0.25mW}$: 50.82 ± 4.8%; unpaired t-test, t=3.697, df = 10, CI = 9.176–37.02, p=0.0041; *Figure 7—figure supplement 2F*). Together, these findings show that the nigral increase in locomotion is not mediated through the PPN, and that dopaminergic input from the SNc to the PPN is rewarding while total input from the substantia nigra to the PPN is aversive.

## Discussion

In this study, we comprehensively characterize the synaptic strength and impact on neuron action potential firing in molecularly- and regionally-defined PPN subpopulations while stimulating the SNr or GPe using ex vivo electrophysiology and optogenetics (*Figure 7—figure supplement 4*). We also stimulated the SNr or GPe axons in the PPN in vivo, which revealed that the SNr and GPe evoke differential locomotion and valence processing behaviors in non-canonical basal ganglia circuits. Together, these findings show that distinct and selective inhibition of PPN subpopulations by the SNr and GPe can alter behavioral output.

### Region- and cell-type-specific inputs to the PPN

The SNr and GPe have been previously shown to form monosynaptic synapses with PPN neurons using rabies viral tracing (*McElvain et al., 2021*; *Zhao et al., 2022*; *Roseberry et al., 2016*; *Huerta-Ocampo et al., 2021*; *Dautan et al., 2021*; *Caggiano et al., 2018*). While SNr input to cholinergic and glutamatergic PPN neurons has been reproduced across different studies (*Zhao et al., 2022*; *Roseberry et al., 2016*; *Huerta-Ocampo et al., 2021*; *Dautan et al., 2021*; *Caggiano et al., 2018*), SNr inputs to GABAergic PPN neurons have only been inferred from studies of the entire mesencephalic locomotor region (MLR), consisting of both the PPN and cuneiform nucleus (*Roseberry et al., 2016*). Several studies have used slice electrophysiology to show that the SNr inhibits PPN neurons generally (*Hormigo et al., 2019*) or to compare SNr input to cholinergic and non-cholinergic PPN neurons (*Takakusaki et al., 1997*; *Granata and Kitai, 1991*; *Kang and Kitai, 1990*). However, a full characterization of the SNr input to regionally and molecularly defined PPN neurons in adult animals has been lacking. Using ex vivo whole-cell patch recordings, we were able to measure inhibitory currents in each of the PPN cell types during SNr stimulation, showing that the SNr functionally inhibits each cell type within the rostral PPN to a similar extent while most strongly inhibiting a medial-caudal 'hotspot' of glutamatergic neurons.

While the GPe has been shown to project to the cholinergic and glutamatergic PPN neurons (*Huerta-Ocampo et al., 2021*; *Dautan et al., 2021*; *Caggiano et al., 2018*), our electrophysiological results show that GPe inhibition of the cholinergic neurons is weak to non-existent and its inhibition of the glutamatergic PPN neurons is caudally biased. In contrast to a previous study showing that the GPe does not project to GABAergic MLR neurons (*Roseberry et al., 2016*), we find that the GPe also preferentially inhibits a caudal subgroup of GABAergic PPN neurons. While the GPe selectively inhibits a caudal subpopulation of GABAergic and glutamatergic PPN neurons, its inhibition is much weaker than SNr inhibition of the PPN in terms of the proportion of connected PPN neurons, the inhibitory current amplitudes, and the inhibition of neuronal firing. Overall, our findings utilize a systematic approach to characterize the synaptic strength between these inhibitory basal ganglia nuclei and the PPN across its rostrocaudal axis for each cell type, allowing us to compare SNr and GPe to PPN circuitry with regional-level granularity. These results also encourage future work involving the PPN to consider the individual influence of its rostral and caudal regions.

### Noncanonical GPe motor circuits

We found that stimulation of GPe axons over the PPN decreased locomotion in both low (putatively local) and high (putatively generating antidromic activity) laser power stimulations. This behavioral outcome appears counter to the canonical model of basal ganglia movement pathways, in which GPe activation increases movement. Although there is strong support for this canonical model, recent in vivo recordings have found subpopulations of GPe neurons that show activity patterns counter to this

model (neurons that increase activity during immobility) (*Dodson et al., 2015*; *Turner and Anderson, 1997*). Similarly, a subset of GPe neurons paradoxically increases activity upon indirect pathway striatal projection neuron stimulation (*Aristieta et al., 2021*). While one explanation for these heterogeneous responses could be local inhibition within the GPe (*Cui et al., 2021*; *Aristieta et al., 2021*; *Kita and Kitai, 1994*; *Sadek et al., 2007*), another possibility is that distinct subpopulations within the GPe differentially modulate locomotor behavior (*Pamukcu et al., 2020*; *Cui et al., 2021*; *Aristieta et al., 2021*; *Dong et al., 2021*; *Mastro et al., 2017*). Our findings support non-canonical basal ganglia motor pathways involving GPe neurons that project to the PPN.

Our characterization of region-specific inhibitory inputs to the PPN helps us understand how GPe inhibition of the PPN can mediate decreases in locomotion. Our electrophysiology experiments show that the GPe exerts minimal influence on the rostral PPN neurons which appear to decrease locomotion (*Goñi-Erro et al., 2023*; *Gut et al., 2022*; *Huang et al., 2024*) and preferentially inhibits caudal GABAergic and glutamatergic PPN neurons. Because stimulating the caudal GABAergic and glutamatergic PPN neurons increases locomotion (*Masini and Kiehn, 2022*), GPe selective inhibition of these caudal neurons could explain our behavioral experiments showing that stimulation of GPe axons over the PPN decreases locomotion. Future studies are needed to determine if GPe inhibition of both cell types is necessary for mediating decreased locomotion. Aligned with our findings, selectively stimulating the Npas1 +or FoxP2 +subpopulations in the GPe decreases locomotion (*Pamukcu et al., 2020*; *Aristieta et al., 2021*). Some groups have found that direct stimulation of the PV +GPe neurons increases locomotion (*Pamukcu et al., 2020*; *Lilascharoen et al., 2021*); however, one study shows that inhibition of PV +GPe neurons can increase locomotion (*Isett et al., 2023*). While we show that GPe projections to the PPN decrease locomotion, future studies are needed to determine which GPe subpopulations project to the PPN to modulate locomotion.

## GABAergic vs dopaminergic effects of nigral projections to the PPN

While we found that stimulating nigral axons in the PPN with high laser power increased locomotion, we observed no locomotor effect when using low laser power stimulation. The simultaneous inhibition of caudal PPN neurons that promote movement (*Masini and Kiehn, 2022*) and rostral PPN neurons that suppress movement (*Goñi-Erro et al., 2023*; *Gut et al., 2022*; *Huang et al., 2024*) may explain a net zero effect on locomotion during low power optical stimulation of SNr axons in the PPN. Indeed, low laser power stimulation of nigral axons in the most rostral implant location resulted in increased locomotion, while the most caudal implant location resulted in decreased locomotion (*Figure 7—figure supplement 3*). Interestingly, the enhanced locomotion observed during high laser power stimulation suggests that a subset of nigral neurons projecting to the PPN promote movement. However, the SNr canonically suppresses movement and the direct somatic stimulation of PV +and GAD2 +SNr neurons that send axon collaterals to the PPN (*McElvain et al., 2021*; *Liu et al., 2020*) decreases gross locomotor movement (*Rizzi and Tan, 2019*; *Liu et al., 2020*). Since injecting constitutive AAVs into the substantia nigra can infect both GABAergic neurons of the SNr and adjacent dopaminergic neurons of the SNc, it is possible that stimulation of dopaminergic activity may mask the impact of stimulating the GABAergic SNr alone. Both the increased locomotion with high laser power and the lack of change with low laser power was recapitulated when we ran these experiments while selectively stimulating dopaminergic projections from the SNc to the PPN (*Figure 7—figure supplement 2*). Therefore, our findings suggest that activation of the PPN-projecting dopaminergic SNc neurons promotes locomotion, but not through direct actions at the PPN. This is consistent with previous literature showing that SNc dopaminergic neurons send axons to both the PPN and striatum (*Ryczko et al., 2016*) and that direct stimulation of nigrostriatal dopaminergic neurons can increase movement initiation and vigor (*da Silva et al., 2018*). Therefore, antidromic stimulation of the SNc may evoke dopamine release into the striatum to increase movement in our high laser power experiments. Future studies are needed to evaluate the motor behavioral impact of the different SNr cell types when inhibiting the entire PPN versus the rostral or caudal PPN subregions.

By contrast, we find that both high and low power non-selective SNr axon stimulation in the PPN is aversive (*Figure 7E–G*), while both high and low power selective stimulation of dopaminergic SNc axons in the PPN is rewarding (*Figure 7—figure supplement 2*). These findings are particularly interesting because they indicate that the dopaminergic vs GABAergic inputs from the substantia nigra to the PPN serve opposite functions regarding valence signaling. Importantly, these findings indicate

that GABAergic input from the SNr is highly aversive and that this aversive signaling occurs through the PPN. Interestingly, these findings also show that dopaminergic SNc neurons do not require antidromic stimulation to signal reward. This means that dopaminergic signaling within the PPN may serve to reinforce behavior. Future work is needed to parse out specific reward and aversion pathways involving neural subpopulations within the SNr, SNc, and PPN.

## Differential PPN inhibition by the SNr and GPe mediates opposing valence processing outcomes

We found that stimulation of SNr axons over the PPN was aversive in real-time place preference (RTPP), while stimulation of GPe axons over the PPN led to a preference for the stimulated chamber. Previous studies directly stimulating SNr and GPe subpopulations have implicated these structures in reward and aversion processing. Inhibition of Vgat +SNr neurons has been shown to increase place preference (*Galaj et al., 2020*) while its direct stimulation has no effect (*Rizzi and Tan, 2019*; *Galaj et al., 2020*). However, selective stimulation of the PV +SNr neurons induces aversion in RTPP (*Rizzi and Tan, 2019*). Although the GPe has been predominantly studied in the context of movement, recent work has begun to reveal a role for the GPe in valence processing (*Isett et al., 2023*; *Baker et al., 2023*; *Farries et al., 2023*; *Arkadir et al., 2004*). One recent study shows that inhibition of PV +GPe neurons induced aversion while inhibition of Npas1 +GPe neurons induced place preference in RTPP (*Isett et al., 2023*). Because activation of PV +SNr neurons is aversive (*Rizzi and Tan, 2019*) and inhibition of PV +GPe neurons is aversive (*Isett et al., 2023*), it is likely that our results are due to activation of the PV +SNr and GPe axons in the PPN. While we show that activation of the PPN-projecting SNr and GPe subpopulations strongly affect valence processing, future studies are needed to determine whether these effects are specifically mediated by PV +SNr and GPe projections to the PPN.

Neurons in the PPN have been implicated in reward and aversive-related behaviors (*Yoo et al., 2017*; *Xiao et al., 2016*; *Dautan et al., 2016*; *Zhang et al., 2024*; *Hormigo et al., 2019*; *Hormigo et al., 2021*; *Liu et al., 2023*; *Aitta-aho et al., 2018*). The cholinergic PPN neurons are implicated in positive valence with place preference and increased lever pressing through their projections to the ventral tegmental area (VTA) (*Xiao et al., 2016*; *Dautan et al., 2016*). Aligned with our findings, direct inhibition of the cholinergic PPN neurons evokes place aversion (*Xiao et al., 2016*). Of particular significance, we show that the SNr inhibits the cholinergic PPN neurons while the GPe axons avoid cholinergic PPN neurons. Therefore, it is possible that the difference between the SNr and GPe effects on valence is due to the difference in their inhibition of PPN cholinergic neurons.

Direct stimulation of the glutamatergic PPN neurons has also been implicated in positive valence with increased reinforcement behavior (*Yoo et al., 2017*). Both the cholinergic and glutamatergic PPN neurons directly influence dopamine release in the striatum and dopaminergic neuron activity in the SNc and VTA (*Yoo et al., 2017*; *Xiao et al., 2016*; *Dautan et al., 2016*; *Forster and Blaha, 2003*; *Estakhr et al., 2017*; *Galtieri et al., 2017*; *Futami et al., 1995*; *Good and Lupica, 2009*). Specifically, the caudal PPN neurons are thought to innervate the medial part of the SNc and the VTA (*Martinez-Gonzalez et al., 2011*; *Good and Lupica, 2009*), two major reward pathway hubs. Our data show that the SNr most strongly inhibits the caudal glutamatergic PPN neurons. Therefore, strong SNr inhibition of these neurons would remove excitatory drive from the reward-related dopaminergic neurons. This pathway could also contribute to the aversive effect of SNr axon stimulation in the PPN. Future experiments are needed to test whether direct inhibition of the caudal glutamatergic PPN neurons receiving SNr input is sufficient to evoke aversion.

The pathways underlying the GPe-mediated rewarding effect are less clear and may indicate functional heterogeneity among caudal PPN neurons. We find that the GPe only inhibits a subset of caudal GABAergic and glutamatergic PPN neurons. It is possible that the subset of GABAergic PPN neurons targeted by the GPe disinhibits local cholinergic and glutamatergic neurons to increase excitation of the rewarding dopaminergic neuron pathways. Another possibility is that the specific GPe-inhibited PPN population may have particularly aversive properties. For example, PPN neurons that excite amygdala nuclei evoke place avoidance (*Liu et al., 2023*; *Aitta-aho et al., 2018*). In this case, GPe inhibition of these neurons could be rewarding by removing excitation of amygdala nuclei involved in negative valence processing. To understand the PPN circuitry involved in GPe-mediated place preference, future experimental work must determine whether inhibiting

GPe-targeted GABAergic or glutamatergic PPN subpopulations alone is sufficient to evoke place preference.

## Conclusions

We have systematically characterized SNr and GPe inputs across the rostrocaudal axis of the PPN for each cell type – cholinergic, GABAergic, and glutamatergic. We show that the SNr inhibits nearly every PPN cell recorded with differential strength determined by cell type and anatomical biases while identifying a medial-caudal 'hotspot' of glutamatergic neurons most strongly inhibited by the SNr. In contrast, the GPe strikingly avoids the cholinergic PPN neurons and weakly, but selectively, inhibits a subpopulation of caudal GABAergic and glutamatergic neurons. We proposed that the differential inhibition of regionally biased, mixed-cell type PPN subpopulations can alter behavioral outputs. By stimulating these inhibitory basal ganglia axons over the PPN, we show that the SNr evokes place aversion while the GPe evokes place preference and decreases locomotion. Together, our findings show that the SNr and GPe mediate opposing valence processing outcomes through the PPN and support a non-canonical basal ganglia motor pathway in which the GPe decreases locomotion through its projections to the PPN.

## Methods

**Key resources table**

| Reagent type (species) or resource | Designation | Source or reference | Identifiers | Additional information |
|---|---|---|---|---|
| Strain, strain background (*Mus musculus*) | *ChAT*-cre mice: B6.129S-*Chat*$^{tm1(cre)Lowl}$/MwarJ | Jackson Laboratory | RRID:IMSR_JAX:031661 | |
| Strain, strain background (*Mus musculus*) | *Vgat*-cre mice: B6J.129S6(FVB)-*Slc32a1*$^{tm2(cre)Lowl}$/MwarJ | Jackson Laboratory | RRID:IMSR_JAX:028862 | |
| Strain, strain background (*Mus musculus*) | *Vglut2*-cre mice: B6J.129S6(FVB)-*Slc17a6*$^{tm2(cre)Lowl}$/MwarJ | Jackson Laboratory | RRID:IMSR_JAX:028863 | |
| Strain, strain background (*Mus musculus*) | Ai9-tdTomato mice: B6.Cg-*Gt(ROSA)26Sor*$^{tm9(CAG-tdTomato)Hze}$/J | Jackson Laboratory | RRID:IMSR_JAX:007909 | |
| Strain, strain background (*Mus musculus*) | *DAT*-cre mice B6.SJL-*Slc6a3*$^{tm1.1(cre)Bkmn}$/J | Jackson Laboratory | RRID:IMSR_JAX006660 | |
| Strain, strain background (*Mus musculus*) | C57BL/6 J | Jackson Laboratory | RRID:IMSR_JAX:000664 | |
| Strain, strain background (*AAV*) | AAV1-hSyn-hChR2(H134R)-EYFP | Addgene (Deisseroth Lab) | Cat#: 26973 | |
| Strain, strain background (*AAV*) | AAV1-Ef1a-double floxed-hChR2(H134R)-EYFP-WPRE-HGHpA | Addgene (Deisseroth Lab) | Cat#: 20298 | |
| Strain, strain background (*AAV*) | AAV1-hSyn-EGFP | Addgene (Roth Lab) | Cat#: 50465 | |
| Antibody | goat polyclonal anti-choline acetyltransferase (ChAT) | Millipore | Cat#: AB144P, RRID:AB_2079751 | (1:200) |
| Antibody | Streptavidin, Cy5 | Invitrogen | Cat#: SA1011 | (1:1000) |
| Antibody | Streptavidin, DyLight 405 | Invitrogen | Cat#: 21831 | (1:1000) |
| Antibody | donkey polyclonal anti-goat IgG (H+L) Cross-Adsorbed Secondary Antibody, Alexa Fluor 647 | Invitrogen | Cat#: A-21447, RRID:AB_2535864 | (1:333) |
| Antibody | sheep polyclonal anti-tyrosine hydroxylase (TH) | Novus Biologicals | Cat# NB300-110, RRID:AB_10002491 | (1:1000) |
| Antibody | DyLight 405 AffiniPure donkey polyclonal anti-Sheep IgG (H+L) | Jackson ImmunoResearch Laboratories | Cat# 713-475-147, RRID:AB_2340740 | (1:100) |
| Chemical compound, drug | D-AP5 | Tocris and HelloBio | Cat#: 0106 and HB0225 | (50 µM) |

*Continued on next page*

*Continued*

| Reagent type (species) or resource | Designation | Source or reference | Identifiers | Additional information |
|---|---|---|---|---|
| Chemical compound, drug | NBQX | Tocris and HelloBio | Cat#: 1044 and HB0443 | (5 μM) |
| Chemical compound, drug | CNQX disodium salt | HelloBio | Cat#: HB0205 | (20 μM) |
| Chemical compound, drug | SR 95531 hydrobromide (GABAzine) | Tocris and HelloBio | Cat#: 1262 and HB0901 | (10 μM) |
| Software, algorithm | Igor Pro | WaveMetrics | | Version 9.00 |
| Software, algorithm | GraphPad Prism | Dotmatics | | Version 9.5.0 |
| Software, algorithm | Fiji software | Open source on GitHub | | Version 1.51 n |
| Software, algorithm | ANY-maze software | Stoelting Company, Wood Dale, IL. | | Version 7.36 |
| Software, algorithm | Clampex software | Molecular Devices | | Version 11.2 |
| Other | Multiclamp amplifier for ex vivo recordings | Molecular Devices | Multiclamp 700B amplifier | |
| Other | Digitizer for ex vivo recordings | Molecular Devices | Axon Digidata 1550B | |
| Other | Confocal microscope | Leica Microsystems | Leica SP8AOBS++ | |
| Other | Fluorescence Microscope | ZEISS | Zeiss Axio Imager Z2 | |
| Other | Horizontal puller for glass microelectrodes for ex vivo recordings | Sutter Instrument Company | Sutter Instrument Model P-97 | |

## Resource availability

Further information and requests for resources and reagents should be directed to and will be fulfilled by the lead contact, Dr. Rebekah C. Evans (re285@georgetown.edu).

## Materials availability

This study did not generate new unique reagents.

## Experimental model and study participant details

### Animal welfare

All animal procedures were approved by the Georgetown University Medical Center Institutional Animal Care and Use Committee (IACUC) (Evans 2020–052). Measures were taken to ensure minimal animal suffering and discomfort, and protocols were designed to minimize the number of animals used.

### Animal subjects

Homozygous *Ai9-TdTomato* mice [B6.Cg-Gt(ROSA)26Sor^tm9(CAG-tdTomato)Hze/J; JAX# 007909] were bred with homozygous *ChAT-Cre* mice [B6.129S-*Chat*^tm1(cre)Lowl/MwarJ; JAX# 031661], homozygous *Vgat-Cre* mice [B6J.129S6(FVB)-*Slc32a1*^tm2(cre)Lowl/MwarJ; JAX# 028862], and *Vglut2-Cre* mice [B6J.129S6(FVB)-*Slc17a6*^tm2(cre)Lowl/MwarJ; JAX# 028863] to obtain *ChAT-Cre/Ai9-tdTomato*, *Vgat-Cre/Ai9-tdTomato*, and *Vglut2-Cre/Ai9-tdTomato* in-house, respectively. *C57BL/6* J wild-type mice (Jax# 000664) and *DAT-cre* mice [B6.SJL-*Slc6a3*^tm1.1(cre)Bkmn/J; JAX#006660] were obtained from Jackson for behavioral experiments. Animals were housed under a 12:12 light-dark cycle (lights on at 06:00 AM) with food and water ad libitum.

### Experimental groups

For electrophysiological studies, *ChAT-Cre/Ai9-tdTomato*, *Vgat-Cre/Ai9-tdTomato*, and *Vglut2-Cre/Ai9-tdTomato* mice of both sexes (3 males and 3 females in each group) aged 2–5 months were used. For behavioral studies, *C57BL/6* J wild-type mice of both sexes, with age-matched littermates

randomly assigned to control or experimental groups. *DAT-cre* mice of both sexes were used for behavioral experiments where specified. Behavioral assessments were exclusively conducted during the light phase, and all testing chambers were cleaned with 70% ethanol between mice to mitigate potential olfactory influences.

## Method details

### Viral injections and optical fiber implantation surgeries

Mice at least 7 weeks old were briefly anesthetized with inhaled 5% isoflurane using an anesthetic chamber and placed onto a heated pad within the stereotaxic frame (Stoelting 51730UD). The skull was stabilized with evenly positioned ear bars and nose cone properly positioned to deliver continuous 1–3% isoflurane and oxygen at a steady flow of 1 L/min throughout the surgery duration. Bupivacaine (5 mg/kg) and carprofen (5 mg/kg) were administered as local anesthetic and analgesic, respectively. A small incision was made on the scalp to visualize bregma and lambda, which were used as references to level the skull. Bilateral holes were drilled at either the coordinates of the SNr (AP –3.1 mm, ML ±1.4 mm, DV –4.7 mm relative to bregma) or GPe (AP –0.3 mm, ML ±1.9 mm, DV –3.9 relative to bregma). The 5 µL Hamilton microsyringe was positioned and 250 nL virus was injected at a rate of 0.2 µL/min. The syringe was raised and rested 0.5 mm above the injection site for 10 min. For ex vivo optogenetic experiments, AAV1-hSyn-hChR2(H134R)-EYFP (2.3x10^13 particles per milliliter, Addgene, Cat# 26973) was injected into either the SNr or GPe. For in vivo optogenetic experiments, AAV1-hSyn-hChR2(H134R)-EYFP 2.0x10^13 particles per milliliter, Addgene, Cat #26973, AAV1-hSyn-EGFP (1.1x10^13 particles per milliliter, Addgene, Cat# 50465), or AAV1-Ef1a-double floxed-hChR2(H134R)-EYFP-WPRE-HGHpA (2.3x10$^{13}$ particles per milliliter, Addgene, Cat#20298) was injected into either the SNr or GPe for experimental and control mice, respectively. For mice used in the in vivo optogenetic experiments, an optical fiber (200 µm core, 0.22 NA, 3.7 mm length) was implanted over the middle-to-rostral PPN (AP –4.5, ML ±1.1 mm, DV –3.5 relative to bregma). The skin on either side of the incision site was joined, glued together using VetBond tissue adhesive, and fastened with two wound clips or the optical fiber was fixed in position with dental cement. (C&B-Metabond Quick! Adhesive Luting Cement by Parkell Products Inc, Patterson# 553–3484/CAT# S380; Jet Denture Repair Package by LANG, CAT# 1223F2). Post-surgery, buprenorphine SR (1.5 mg/kg) was administered for long-acting analgesia. Mice were allowed to recover on a heating pad until fully awake and were monitored daily for signs of distress or infection.

### Electrophysiological solutions

Acute brain slices were prepared in modified artificial cerebrospinal fluid (aCSF) solutions with final osmolarities ~300–310 mOsm and ~7.4 pH. The slicing solution contained (in mM) 198 glycerol, 2.5 KCl, 1.2 NaH$_2$PO$_4$, 20 HEPES, 25 NaHCO$_3$, 10 glucose, 10 MgCl$_2$, and 0.5 CaCl$_2$. The holding solution contained (in mM) 92 NaCl, 2.5 KCl, 1.2 NaH$_2$PO$_4$, 30 NaHCO$_3$, 20 HEPES, 2 MgCl$_2$, 2 CaCl$_2$, 35 glucose, 5 sodium ascorbate, 3 sodium pyruvate, and 2 thiourea. Recording aCSF was made up of (in mM): 125 NaCl, 25 NaHCO$_3$, 3.5 KCl, 1.25 NaH$_2$PO$_4$, 10 glucose, 1 MgCl$_2$, 2 CaCl$_2$. Whole-cell patch clamp recordings used a potassium methane sulfonate (KMeSO$_3$)-based internal solution containing (in mM) 122 methanesulfonic acid, 9 NaCl, 9 HEPES, 1.8 MgCl$_2$, 4 Mg-ATP, 0.3 Tris-GTP, and 14 phosphocreatine for a final osmolarity between 290 and 305 mOsm. The internal solution contained neurobiotin (0.1–0.3%) for post-hoc staining. These reagents enable reliable electrophysiology recordings in adult brainstem neurons (*Chen and Evans, 2024*; *Evans et al., 2020*).

### Slicing and electrophysiology

Animals ages 2–5 months old were anesthetized with inhaled isoflurane and transcardially perfused with ice-cold slicing solution that had been bubbled with 95% O$_2$ and 5% CO$_2$. Mice were decapitated and brains were quickly extracted from the skull, keeping the cerebellum intact. For sagittal slices, the brain hemispheres were separated and the medial side was glued onto a 3% agar block fixed to the stage of a semi-automatic Leica VT1200 microtome. 200 µm-thick slices were obtained and incubated at 34 °C in holding solution for 30 min then kept at room temperature. In all steps, the modified aCSF solutions are bubbled with 95% O$_2$ and 5% CO$_2$. Slices in the recording chamber during whole-cell patch clamp experiments were continuously perfused with oxygenated recording aCSF kept at 28–34°C using a water bath and in-line Warner heater. Cells were visualized using an Olympus

OpenStand upright microscope and 565 nm ThorLabs LED light. Recording pipettes with resistance between 1.5 and 4 MΩ were prepared using borosilicate glass capillaries (World Precision Instruments, Inc 1B150F-4, 4-inch length, 1.5 mm OD, 0.84 mm ID) with a micropipette puller (Sutter Instrument Model P-97). Recordings were obtained using a Multiclamp 700B amplifier and Axon Digidata 1550B controlled by Clampex 11.2. Voltage-clamp signals were low-pass filtered at 2 kHz and sampled at 10 kHz. Current-clamp signals were low-pass filtered at 10 kHz and sampled at 10 kHz. At the end of recording, the cell was sealed by moving the recording pipette slightly above the soma and an image was taken of the PPN and pipette tip.

## Ex vivo optogenetic activation

Whole-field optogenetic activation of channelrhodopsin-infected axons in brain slice was achieved with a blue (470 nm) ThorLabs LED light sent to the tissue via a silver mirror. Light intensity measured at the objective back aperture was 13 mW. Light activation was applied at 20 Hz with 2ms pulse intervals for 2 s. All recordings were conducted in the presence of glutamatergic receptor blockers – 50 µM D-AP5 (Tocris Cat#0106 and HelloBio Cat#H0225), and 5 µM NBQX (Tocris Cat#1044 and HelloBio Cat#HB0443) alone or in combination with 20 µM CNQX (HelloBio Cat#HB0205).

In voltage-clamp, cells were determined to be connected if observable optically evoked inhibitory postsynaptic currents (oIPSCs) were measured while holding the cell at –50 mV. In a subset of neurons, 10 µM GABAzine (Tocris Cat#1262 and HelloBio Cat#HB0901) was applied to ensure the oIPSCs were GABA-A receptor mediated. In a few cells (17/54), small inhibitory currents remained with amplitudes averaging about 24% of the first oIPSC measured. Any cells that required greater than 200 pA to be held at –50 mV or that had access resistance that exceeded 30 MΩ at the start of recording were excluded from oIPSC quantitative analyses as they were likely unhealthy and too leaky; however, the first oIPSC amplitude was used as an approximate measure of input strength in the cell maps (SNr: 21/46; GPe: 11/40). oIPSC amplitudes were measured from the baseline to the peak of each current. The paired pulse ratio (PPR) was calculated from the amplitudes of the first two currents evoked in the 20 Hz train.

For current-clamp recordings, most patched PPN neurons spontaneously fired, but a few required a small amount of hyperpolarizing or depolarizing current to fire (R)ostral, (C)audal; SNr-PPN$_{ChAT}$ R: 3/14, C: 4/23; SNr-PPN$_{Vgat}$ R: 2/7, C: 9/19; SNr-PPN$_{Vglut2}$ R: 2/13, C: 1/17; GPe-PPN$_{ChAT}$ R: 2/7, C: 9/25; GPe-PPN$_{Vgat}$ R: 2/7, C: 8/18; GPe-PPN$_{Vglut2}$ R: 6/12, C: 7/29 required current. Neurons injected with more than 60 pA were excluded. In this manuscript, we use 'pre-stimulation firing frequency' to refer to the time period right before optical stimulation. The pre-stimulation firing frequency was measured using the number of action potentials in the 1-s epoch before the start of stimulation. The firing frequency during stimulation was measured using the middle one-second of the 2 s stimulation. The post-stimulation firing frequency was measured in the 1-s epoch immediately following the end of the stimulation. The absolute frequency change is the difference between the pre-simulation and during stimulation frequencies. The rebound frequency change is the difference between the post-stimulation and pre-stimulation frequencies. These values include cells with and without holding current applied. By contrast, the spontaneous firing rate of the cell includes only cells with no holding current applied.

## In vivo optogenetic activation

Two behavioral tests, the open field (OF) test and RTPP, were conducted sequentially to assess gross locomotion and place preference in animals of at least 11 weeks old (mean age ± SEM; SNr-injected mice: 132±23.4 days old; GPe-injected mice: 110±7.4 days old). Optical stimulation was achieved by the application of blue light (473 nm) at 20 Hz with 2ms pulses (4% duty cycle). Behavioral testing was conducted under both high laser power (4.0–4.5 mW) and low laser power (0.2–0.25 mW) conditions. Low power testing was employed to control for potential behavioral effects related to antidromic stimulation. On day 1, mice underwent OF testing and were randomly assigned to high or low power stimulation conditions. On day 2, mice were tested in the open field arena for a second time under opposite power conditions. On day 3, mice were tested in the RTPP arena for 10 min and were again randomly assigned to high or low power conditions. On day 4, mice were tested in the RTPP arena for 10 min without any stimulation to assess retention of place memory. Finally, on day 5, mice

underwent RTPP under opposite power conditions (low vs high) and opposite chamber as stimulation zone (stripes vs spots) to those on Day 3.

## Open field test

OF test was conducted in an opaque arena measuring 40.64x40.64 x 40.64 cm. Animals were placed in the center of the arena and allowed to explore freely for 10 min. This was followed by 5 min of discontinuous bilateral photostimulation at either high or low power, 5 min of recovery, another 5 min of discontinuous photostimulation at 20 Hz (at either high or low power), and a final 5 min of recovery. The discontinuous photostimulation consisted of three 1 min photostimulation periods interspersed with 1 min intervals of no photostimulation. Movement during the OF test was recorded and analyzed using ANY-maze software (Stoelting Company, Wood Dale, IL).

## Real-time place preference

The real-time place preference (RTPP) apparatus consisted of a rectangular behavioral arena with three chambers, each measuring 24.60x27.94 x 27.94 cm, with a non-reflective gray background. The adjacent chambers had distinct visual cues: one chamber featured stripes and the other spots. These striped and spotted zones were randomly assigned as stimulation zones between animals. The center chamber and one of the adjacent chambers served as neutral zones, while the other adjacent chamber served as a stimulation zone. Animals received unilateral optical stimulation when in the assigned stimulation zone. The laser-on side was randomly chosen and counter-balanced between mice. Mice were also randomly assigned to have low laser power RTPP first or high laser power RTPP first. In each case, mice were given an unstimulated 10 min trial on the day between the first and second RTPP experiment to 'unlearn' which side was stimulated, and the second RTPP experiment stimulated the opposite chamber compared to the first RTPP experiment. For example, one mouse would have high power stimulation on the striped side on day 1, no stimulation on day 2, and low power stimulation on the spotted side on day 3. To mitigate locomotor effects with bilateral stimulation, unilateral optical stimulation was used in RTPP. Individual subjects were placed in the RTPP apparatus and allowed 10 min to explore both compartments. The time spent in each compartment was recorded in real-time using ANY-maze software (Stoelting Company, Wood Dale, IL).

## Immunohistochemistry and confocal imaging

After electrophysiological experiments, brain slices were fixed overnight in a 4% w/v paraformaldehyde (PFA) solution in phosphate buffer (PB) solution, pH 7.6 at 4 °C. The fixed brain slices were then stored in phosphate buffer (PB) solution until immunostaining.

Brain slices were also collected at the conclusion of behavioral experiments. Mice were deeply anesthetized and perfused with phosphate buffer (PB), followed by a fixative solution containing 4% w/v paraformaldehyde (PFA) in PB, pH 7.6, at 4 °C. Whole brains were extracted and fixed overnight in the same PFA solution. After fixation, brains were stored in PB solution until further processing. For cubic processing, 200 µm-thick sagittal slices were obtained from post-behavioral whole mouse brains using the PELCO easiSlicer Vibratory Tissue Slicer.

A CUBIC tissue clearing protocol (*Susaki et al., 2015*) was combined with immunofluorescence staining as in *Evans et al., 2020* for all fixed brain slices from both electrophysiological and behavioral experiments. All steps are performed at room temperature on a shaker. Slices were placed in CUBIC reagent 1 for 1–2 days; then washed in PB 3 x for 1 hr each; placed in blocking solution (0.5% fish gelatin in PB) for 3 hr; placed in primary antibodies for 2–3 days; washed in PB 3 x for 2 hr each; placed in secondary antibodies for 2–3 days; washed in PB 3 x for 2 hr each; and placed in CUBIC reagent 2 for 2 hr before mounting onto slides (Fisherbrand 12-550-403) in reagent 2 and sealed with frame-seal incubation chambers (Thermo Scientific AB-0577) and a coverslip (Corning 2845–18).

Neurobiotin-filled patched neurons were stained with streptavidin antibody (Cy5, Invitrogen Cat#SA1011, 1:1000 or DyLight 405, Invitrogen Cat#21831, 1:1000). Goat anti-ChAT primary antibody (Sigma Cat#AB144P, 1:200) and donkey anti-goat Alexa Fluor 647 secondary antibody (Invitrogen Cat#A-21447, 1:333) was used to identify the borders of the PPN. On an example slice to show the SNr injection site, sheep anti-TH primary antibody (Novus Biologicals Cat#NB300-110, 1:1000) and donkey anti-sheep (1:100) were used to delineate the GABAergic SNr from the dopamine-rich TH +SNc neurons. To identify the approximate location of patched cells for cell mapping, slices were

imaged as tiled z-stacks using a Leica SP8AOBS ++at the Microscopy and Imaging Shared Resource core facility at Georgetown University. Additionally, to verify the injection site and optical implant placement in the brain slices from mice that underwent behavioral testing, tiled z-stacks were obtained using either the aforementioned Leica SP8AOBS ++or a Zeiss Axio Imager Z2 at the Georgetown Department of Neuroscience core imaging facility.

## Approximate cell location

An image of the PPN with the pipette tip still in position was taken at the end of each cell recording and confocal images of the slices after CUBIC clearing were obtained as described above. Confocal images were overlapped with the pipette images using the streptavidin-stained patched cells (which aligned with the pipette tip images) and autofluorescence of the superior cerebellar peduncle (scp) white fibers as landmarks. The perimeter of the PPN was determined by the presence of ChAT +neurons and generally extended from the edge of the SNr to approximately 2/3 up the scp fibers. ChAT +neurons were identified by tdTomato expression in the ChAT-Cre mouse line and anti-ChAT staining in the Vgat-Cre and Vglut2-Cre mouse lines. Since there is variability introduced during the slicing process, we matched slices to three sagittal templates representing 1.00, 1.25, and 1.50 mm lateral to the midline in the Paxinos and Franklin's Mouse Brain Atlas 4th edition. The most medial slice (1.00 mm lateral to the midline) characteristically had few to no fibers of the superior cerebellar peduncle at the edge of the slice where the cerebellum would tether. All neurons on this slice were considered 'caudal' following the typical shape of the PPN. Slices matching 1.25 and 1.50 mm from the midline had a region of densely clustered cholinergic neurons and a region of loosely dispersed cholinergic neurons. An oval ROI was created following the length and orientation of ChAT +neuron spread. The midline of the oval demarcated the rostral and caudal regions. Overlapping the midlines allowed us to evaluate rostral-caudal differences across mice with a small margin of error, particularly along the midline. For this reason, we also mapped the location of patched cells, which shows the distance from the midline and can reveal medial-lateral differences. The XY coordinates were determined using Fiji software and plotted in Igor. The color scale applied to the points represented the amplitude of the first oIPSC and ranged from 0 to 250 pA so that amplitudes greater than 250 pA were shown in the same maximal bright color.

## Approximate implant location

To verify the location of the optical implant, anatomical landmarks for locating the PPN were identified by immunofluorescent labeled PPN cholinergic neurons. Sections containing visible optical fiber scarring were stained with anti-ChAT immunofluorescent labeling. These histological images were then visually matched to the corresponding sagittal planes of the stereotaxic atlas, based on the identified anatomical landmarks. The approximate position of the optical cannula was localized in the histological images and was in the middle-to-rostral part of the PPN.

## Quantification and statistical analysis

Electrophysiological traces were processed in Igor Pro 9 (Wavemetrics), and all statistical analyses were performed in GraphPad Prism 9.5. All parametric data in text is reported as mean ± SEM. Two-tailed unpaired t-tests were used to compare between two groups and one-way ANOVA followed by Dunnett's post hoc test was used to compare three or more groups. If the equality of variances was violated (i.e. the ratio of the largest and smallest standard deviations was greater than 2 and Bartlett test was statistically significant), Welch's ANOVA test was alternatively used. Two-way ANOVA followed by Fisher's LSD post hoc test was performed for percent of pre-optical stimulation frequency as each comparison group was predetermined. Non-parametric data in text is reported as median [interquartile range (IQR)]. Boxplots show medians, 25th and 75th percentiles as first and third quartile box edges, and 9th and 91st percentiles as whiskers. We used the Mann-Whitney rank-sum test for comparing two groups and the Kruskal-Wallis test for multiple comparisons. As each comparison group within the Kruskal-Wallis tests was planned and stand-alone, uncorrected Dunn's post hoc test was performed to determine significance between groups. To determine if caudal GPe connectivity is significantly greater than rostral GPe connectivity, as expected from axon projection patterns, a one-tailed Fisher's exact test and odds ratio effect size was performed. For correlation matrix analyses, non-parametric Spearman r was computed. Statistical details of

experiments can be found in the text, figure legends, and *Supplementary file 1*. Biological replicates are individual cells (n) from 6 separate mice (N=3 males and 3 females) in the electrophysiological experiments and are individual mice of both sexes in the behavioral experiments (numbers described in results text). Same-sex littermates were randomly allocated to control or experimental groups, and experimenters were not blinded during experiments. Statistical significance was evaluated as $p < 0.05$.

## Acknowledgements

Funding for this research was provided by an NIH BRAIN Initiative grant R00NS112417 (to RCE), NIH NINDS grant 1F30NS132399 (to MF), Michael J Fox Foundation Research Grant MJFF-022863 (to RCE), and Parkinson's Foundation Summer Student Fellowship PF-SSF-941810 (to AES). This research was supported by the Microscopy and Imaging Shared Resource (MISR, S10RR025661) and the Lombardi Comprehensive Cancer Center grant (P30-CA051008). We would like to thank the Division of Comparative Medicine for their animal care and management and Dr. Stefano Vicini, Dr. Aryn Gittis, and members of the Evans lab for their feedback on earlier versions of this manuscript.

## Additional information

### Funding

| Funder | Grant reference number | Author |
|---|---|---|
| BRAIN Initiative | R00NS112417 | Rebekah C Evans |
| National Institute of Neurological Disorders and Stroke | F30NS132399 | Michel Fallah |
| Parkinson's Foundation | PF-SSF-941810 | Aleksandra E Swiatek |
| Michael J Fox Foundation for Parkinson's Research | MJFF-022863 | Rebekah C Evans |

The funders had no role in study design, data collection and interpretation, or the decision to submit the work for publication.

### Author contributions

Michel Fallah, Conceptualization, Data curation, Formal analysis, Investigation, Methodology, Writing – original draft, Writing – review and editing; Kenea C Udobi, Formal analysis, Investigation; Aleksandra E Swiatek, Chelsea B Scott, Investigation, Methodology; Rebekah C Evans, Conceptualization, Resources, Data curation, Formal analysis, Supervision, Funding acquisition, Project administration, Writing – review and editing

### Author ORCIDs

Michel Fallah (ID) https://orcid.org/0000-0001-9762-4087
Rebekah C Evans (ID) https://orcid.org/0000-0002-9476-150X

### Ethics

All animal procedures were approved by the Georgetown University Medical Center Institutional Animal Care and Use Committee (IACUC) (Evans 2020-052). Measures were taken to ensure minimal animal suffering and discomfort, and protocols were designed to minimize the number of animals used.

Reviewer #1 (Public review): https://doi.org/10.7554/eLife.102308.3.sa1
Reviewer #2 (Public review): https://doi.org/10.7554/eLife.102308.3.sa2
Reviewer #3 (Public review): https://doi.org/10.7554/eLife.102308.3.sa3
Author response https://doi.org/10.7554/eLife.102308.3.sa4

## Additional files

### Supplementary files
Supplementary file 1. Supplemental statistical tables.

MDAR checklist

### Data availability
Source data generated for Figures 2–7 and the figure supplements is available on Dryad (https://doi.org/10.5061/dryad.gf1vhhn1x). No original code was generated in this article.

The following dataset was generated:

| Author(s) | Year | Dataset title | Dataset URL | Database and Identifier |
|---|---|---|---|---|
| Fallah M, Udobi KC, Swiatek AE, Scott CB, Evans RC | 2025 | Inhibitory basal ganglia nuclei differentially innervate pedunculopontine nucleus subpopulations and evoke opposite motor and valence behaviors | https://doi.org/10.5061/dryad.gf1vhhn1x | Dryad Digital Repository, 10.5061/dryad.gf1vhhn1x |

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
