## [Editor Report · eLife Assessment]

Fallah et al carefully dissect projections from substantia nigra pars reticulata (SNr) and the globus pallidus externa (GPe) - two key basal ganglia nuclei - to the pedunculopontine nucleus (PPN), a brainstem nucleus that has a central role in motor control. They consider inputs from these two areas onto three types of downstream PPN neurons - GABAergic, glutamatergic, and cholinergic neurons - and carefully map connectivity along the rostrocaudal axis of the PPN. Overall, this **important** study provides **convincing** data on PPN connectivity with two key input structures that will provide a basis for further understanding PPN function.

---

## [Referee Report · Reviewer #1 (Public review)]

Summary:

Fallah and colleagues characterize the connectivity between two basal ganglia output nuclei, the SNr and GPe, and a the pedunculopontine nucleus, a brainstem nucleus that is part of the mesencephalic locomotor region. Through a series of systematic electrophysiological studies, they find that these regions target and inhibit different populations of neurons, with anatomical organization. Overall, SNr projects to PPN and inhibits all major cell types, while the GPe inhibits glutamatergic and GABAergic PPN neurons, and preferentially in the caudal part of the nucleus. Optogenetic manipulation of these inputs in the had opposing effects on behavior - SNr terminals in the PPN drove place aversion, while GPe terminals drove place preference.

Strengths:

This work is thorough and systematic characterization of a set of relatively understudied circuits. They build on the classic notions of basal ganglia connectivity and suggest a number of interesting future directions to dissect motor control and valence processing in brainstem systems.

Limitations:

All the cell type recording studies showing subtle differences in the degree of inhibition and anatomical organization of that inhibition suggest a complex effect of general optogenetic manipulation of SNr or GPe terminals in the PPN. It will be important to determine if SNr or GPe inputs onto a particular cell type in PPN are more or less critical for the how the locomotion and valence effects demonstrated here.

---

## [Referee Report · Reviewer #2 (Public review)]

Strengths:

Fallah et al carefully dissect projections from SNr and GPe - two key basal ganglia nuclei - to the PPN, an important brainstem nucleus for motor control. They consider inputs from these two areas onto 3 types of downstream PPN neurons: GABAergic, glutamatergic, and cholinergic neurons. They also carefully map connectivity along the rostrocaudal axis of the PPN. They provide important and convincing data on PPN connectivity with two important input structures, which will provide a foundation for many future studies. They also consider the behavioral relevance of these different PPN inputs for controlling movement and reinforcement, showing convincing evidence that SNr and GPe inputs have opposing effects on behavior.

Weaknesses:

The optogenetics and behavioral studies are intriguing, although more work will be required to fit these data together into a specific model of circuit function and to distinguish the locomotor and reinforcement effects. Interestingly, stimulation of SNr axons in the rostral vs caudal PPN likely differs (as predicted by slice experiments), indicating an area for future investigation and dissection of pathways.

---

## [Referee Report · Reviewer #3 (Public review)]

The study by Fallah et al. provides a thorough characterization of the effects of two basal ganglia output pathways, the SNr and the GPe, on cholinergic, glutamatergic, and GABAergic neurons of the PPN. Using a combination of optogenetics-assisted electrophysiology and behavioral assays in genetically defined mouse lines, the authors show that SNr projections broadly inhibit all PPN subtypes along the rostrocaudal axis, whereas GPe projections are mostly restricted to the caudal PPN and predominantly target glutamatergic neurons, with a lesser effect on GABAergic neurons. Activation of these inputs in vivo revealed opposing behavioral effects: SNr stimulation increased locomotion and caused avoidance in the real-time place preference (RTPP) task, while GPe stimulation reduced locomotion and increased time spent in the stimulation zone.

Strengths:

The evidence for functional connectivity between SNr and GPe inputs and specific PPN cell types is solid and highlights a prominent influence of SNr across the PPN. The identification of a GPe projection that selectively targets caudal glutamatergic PPN neurons is unexpected and highly relevant to understanding basal ganglia-brainstem interactions. The study stands out for its systematic cell-type-specific approach and the combination of electrophysiological and behavioral data. Importantly, the authors addressed key concerns from the initial review by performing new analyses and adding important controls:

Motor activity was re-analyzed at higher temporal resolution, revealing more nuanced effects of stimulation (Fig. S2).

The concern that motor effects might confound RTPP performance was mitigated by analyzing unstimulated test sessions, which showed that place preference or aversion persisted in the absence of stimulation (Fig. 7G).

The potential recruitment of SNc dopaminergic projections was directly tested using DAT-Cre mice, confirming that dopaminergic axon stimulation drives locomotion and reward but does not explain the aversive effect seen with broader SNr activation (Fig. S3).

Weaknesses:

While the revised analyses and added data strengthen the conclusions, the interpretation of the behavioral effects remains somewhat limited by the use of RTPP, which can be influenced by motor changes, even with unilateral stimulation. Nonetheless, the additional controls and thorough discussion now acknowledge and address these caveats appropriately.

Some minor clarifying edits would enhance the manuscript's precision and readability, including improvements to terminology, data presentation, figure referencing, and the organization of behavioral and statistical reporting.

Conclusion:

This is a strong and compelling study that provides a detailed and novel characterization of basal ganglia inputs to the PPN and their behavioral relevance. The authors were responsive to reviewer feedback, and the revised manuscript is significantly improved. The findings advance our understanding of how basal ganglia output pathways engage brainstem circuits to modulate locomotion and valence.

---

## [Author Response]

The following is the authors’ response to the original reviews.

**Reviewer #1 (Public review):**
Fallah and colleagues characterize the connectivity between two basal ganglia output nuclei, the SNr and GPe, and the pedunculopontine nucleus, a brainstem nucleus that is part of the mesencephalic locomotor region. Through a series of systematic electrophysiological studies, they find that these regions target and inhibit different populations of neurons, with anatomical organization. Overall, SNr projects to PPN and inhibits all major cell types, while the GPe inhibits glutamatergic and GABAergic PPN neurons, and preferentially in the caudal part of the nucleus. Optogenetic manipulation of these inputs had opposing effects on behavior - SNr terminals in the PPN drove place aversion, while GPe terminals drove place preference.Strengths:This work is a thorough and systematic characterization of a set of relatively understudied circuits. They build on the classic notions of basal ganglia connectivity and suggest a number of interesting future directions to dissect motor control and valence processing in brainstem systems.We thank the reviewer for these positive comments.Weaknesses:Characterization of the behavioral effects of manipulations of these PPN input circuits could be further parsed, for a better understanding of the functional consequences of the connections demonstrated in the ephys analyses.

We have further analyzed our behavioral data to reveal more nuanced functional effects and included these analyses in Figure S2.

All the cell type recording studies showing subtle differences in the degree of inhibition and anatomical organization of that inhibition suggest a complex effect of general optogenetic manipulation of SNr or GPe terminals in the PPN. It will be important to determine if SNr or GPe inputs onto a particular cell type in PPN are more or less critical for how the locomotion and valence effects are demonstrated here.

This is a really interesting future direction and we have expanded on these points in the discussion in lines 771-772 and 782-785.

**Reviewer #1 (Recommendations for the authors):**
(1) Overall these are really valuable studies and help set up a number of future directions.

We thank the reviewer for their positive comments.

(2) I don't have many specific suggestions, but more examples of viral targeting and cell type targeting, including potentially some validation of the genetic identity of the cells targeted, could be useful for considering the details of the ephys experiments.

We agree that understanding which exact SNr and GPe neurons go to which exact PPN populations is an important next step and are planning to conduct future experiments investigating these important questions. Others have found that there is minimal overlap between the three cell types within the PPN discussed in this manuscript (Wang and Morales 2009; Yoo et al. 2017; Steinkellner, Yoo, and Hnasko 2019). One important line of future investigations is to look at the specific inputs onto recently identified subsets of the glutamatergic PPN neurons such as Chx10- and Rbp4-expressing neurons (Goñi-Erro et al. 2023; Ferreira-Pinto et al. 2021). We hope to explore the electrophysiological properties and connectivity of these subtypes in future projects.

(3) More discussion of which PPN cell types might be mediating the optogenetic behavioral effects of bulk SNr or GPe terminal stimulation would be useful for connecting the ephys results with the behavior.

We are also interested in the question of which PPN cell type is most critical for mediating the effects observed in bulk terminal stimulation. While the best experiment would be to stimulate the axons projecting to each specific cell type of the PPN, this is not currently possible due to methodological limitations and lack of studies dissecting which SNr and GPe subpopulations project to each cell type of the PPN. However, in future studies, we plan to leverage the ability of AAV1 to jump a synapse along with Cre/Flp viruses and mouse lines to selectively inhibit cholinergic, GABAergic, or glutamatergic PPN neurons that receive GPe or SNr input to elucidate the contribution of each cell type in mediating behavioral changes in movement and valence processing.

To address these important future directions, we have added additional text in the discussion in lines 771-772 and 782-785.

**Reviewer #2 (Public review):**
Summary:Fallah et al carefully dissect projections from SNr and GPe - two key basal ganglia nuclei - to the PPN, an important brainstem nucleus for motor control. They consider inputs from these two areas onto 3 types of downstream PPN neurons: GABAergic, glutamatergic, and cholinergic neurons. They also carefully map connectivity along the rostrocaudal axis of the PPN.Strengths:The slice electrophysiology work is technically well done and provides useful information for further studies of PPN. The optogenetics and behavioral studies are thought-provoking, showing that SNr and GPe projections to PPN play distinct roles in behavior.

We appreciate the reviewer’s positive evaluation.

Weaknesses:Although the optogenetics and behavioral studies are intriguing, they are somewhat difficult to fit together into a specific model of circuit function. Perhaps the authors can work to solidify the connection between these two arms of the work.

We have expanded on these topics in the discussion.

Otherwise, there are a few questions whose answers could add context to the interpretation of these results:(1) Male and female mice are used, but the authors do not discuss any analysis of sex differences. If there are no sex differences, it is still useful to report data disaggregated by sex in addition to pooled data.

We have added a supplementary figure (Figure S2) showing distance traveled during optical stimulation for male and female mice.

(2) There is some lack of clarity in the current manuscript on the ages used - 2-5 months vs "at least 7 weeks." Is 7 weeks the time of virus injection surgery, then recordings 3 weeks later (at least 10 weeks)? Please clarify if these ages apply equally to electrophysiological and behavioral studies. If the age range used for the test is large, it may be useful to analyze and report if there are age-related effects.

Thanks for pointing this out, we have clarified this in the methods. 7 weeks is the youngest age at which mice used for electrophysiology were injected, and all were used for electrophysiology between 2-5 months. For behavior, the youngest mice used were 11 weeks old at time of behavior (8 weeks old at injection). Mice in the GPe-stimulated condition were 110 ± 7.4 SEM days old and mice in the SNr-stimulated condition 132 ± 23.4 SEM days old. We have added these details to the revised manuscript in lines 913 and 963-964.

In addition, we have correlated distance traveled at baseline and during stimulation with age for both SN and GPe stimulated conditions. Baseline distance traveled did not correlate with age, but there was a trend toward more movement during stimulation with older mice in the SN axon stimulation group. We have included these plots in supplemental Figure S2.

(3) Were any exclusion criteria applied, e.g. to account for missed injections?

All injection sites and implant sites were within our range of acceptability, so we did not exclude any mice for missed injections or incorrect implant location.

(4) 28-34 degC is a fairly wide range of temperatures for electrophysiological recording, which could affect kinetics.

This is an important consideration, and we agree the wide temperature is not optimal. We have plotted our main measurement of current amplitude in the condition where we found significant differences between rostral and caudal PPN (SNr to Vglut2 PPN neurons) against temperature and found no correlation (Pearson’s r value = -0.0076). Similarly, we found no correlation between baseline (pre-opto) firing frequency and temperature (r = -0.068). See Author response image 1.

**Author response image 1. sa4fig1:** 

(5) It would be good to report the number of mice used for each condition in addition to n=cells. Statistically, it would be preferable not to assume that each cell from the same mouse is an independent measurement and to use a nested ANOVA.

For electrophysiology, the number of mice used in each experiment was 6 (3 male, 3 female). In the manuscript ‘N’ represents number of mice and ‘n’ represents number of cells. Because of the unpredictability of how many healthy cells can be recorded from one mouse, our data were planned to be collected with n=cells, and are underpowered for a nested ANOVA.

However, in many cases, rostral and caudal data were collected from the same mice. While we do not have sufficient paired data for each electrophysiological parameter, analyzing one of our main and most important findings with a paired comparison (with biological replicates being mice) shows a statistically significant difference in the inhibitory effect of SNr axon stimulation on firing rate between rostral and caudal glutamatergic neurons (p=0.031, Wilcoxon signed rank test). See Author response image 2.

**Author response image 2. sa4fig2:** 

**Reviewer #3 (Public review):**
Summary:The study by Fallah et al provides a thorough characterization of the effects of two basal ganglia output pathways on cholinergic, glutamatergic, and GABAergic neurons of the PPN. The authors first found that SNr projections spread over the entire PPN, whereas GPe projections are mostly concentrated in the caudal portion of the nucleus. Then the authors characterized the postsynaptic effects of optogenetically activating these basal ganglia inputs and identified the PPN's cell subtypes using genetically encoded fluorescent reporters. Activation of inputs from the SNr inhibited virtually all PPN neurons. Activation of inputs from the GPe predominantly inhibited glutamatergic neurons in the caudal PPN, and to a lesser extent GABAergic neurons. Finally, the authors tested the effects of activating these inputs on locomotor activity and place preference. SNr activation was found to increase locomotor activity and elicit avoidance of the optogenetic stimulation zone in a real-time place preference task. In contrast, GPe activation reduced locomotion and increased the time in the RTPP stimulation zone.Strengths:The evidence of functional connectivity of SNr and GPe neurons with cholinergic, glutamatergic, and GABAergic PPN neurons is solid and reveals a prominent influence of the SNr over the entire PPN output. In addition, the evidence of a GPe projection that preferentially innervates the caudal glutamatergic PPN is unexpected and highly relevant for basal ganglia function.Opposing effects of two basal ganglia outputs on locomotion and valence through their connectivity with the PPN.Overall, these results provide an unprecedented cell-type-specific characterization of the effects of basal ganglia inputs in the PPN and support the well-established notion of a close relationship between the PPN and the basal ganglia.

We thank the reviewer for their positive comments.

Weaknesses:The behavioral experiments require further analysis as some motor effects could have been averaged out by analyzing long segments.

We have further analyzed our motor effects and included these analyses in supplemental figure S2 in the revised manuscript.

Additional controls are needed to rule out a motor effect in the real-time place preference task.

To address this comment, we analyzed the second day of RTPP, where no stimulation was applied in either chamber. Specifically, we evaluated the time spent in the stimulated chamber during the first minute of the unstimulated RTPP task. We found that the mice that had SNr axon stimulation still avoided the previously stimulated chamber and the mice that had GPe axon stimulation still preferred the previously stimulated chamber. These data have been added to Figure 7 and in the results section lines 564-575.

Importantly, the location of the stimulation is not reported even though this is critical to interpret the behavioral effects.

The implant locations were generally over the middle-to-rostral PPN and we will clarify this in the revised manuscript. These locations are shown in figure 7B.

There are some concerns about the possible recruitment of dopamine neurons in the SNr experiments.

We have added experiments stimulating the SNc dopaminergic neuron axons in the PPN and found very interesting behavioral effects. These are described in more detail below and in the results lines 595-624. These data are also included in Figure S3.

**Reviewer #3 (Recommendations for the authors):**
(1) Locomotor activity should be analyzed as trial averages instead of session averages. The effect of SNr on locomotion might be showing a rebound of activity in cholinergic neurons, which innervate dopamine neurons and induce locomotion. Furthermore, the variability between animals should be reported, Figure 7C doesn't show a standard deviation.

This is an important point and could reveal different early and late effects of basal ganglia axon stimulation. We have added a time course graph of the trial averages for the distance traveled in the open field with higher temporal resolution (10s vs 1min). This is included in supplemental Figure S2A&B.

The variability between animals was shown as shaded area, but was too light and transparent so it was difficult to see in Figure 7C. We have changed this shading to error bars for better visibility.

(2) SNc projects to the PPN. It has recently been shown that PPN neurons respond robustly to dopaminergic activation, including effects on motor activity (Juarez Tello et al., 2024). The transductions shown in Figure S1 clearly cover to entirety of the SNc. Dopamine blockers should be used in the ex vivo experiments to rule out dopaminergic effects.

This is an important point and one we were particularly interested in as far as the behavioral experiments. We thank the reviewer for bringing this up because it led us to a really interesting result. We have now run an additional experiment using DAT-cre mice and a cre-dependent ChR2 using the same injection site at our constitutive ChR2 experiments. We found that selectively stimulating the SNc dopaminergic axons replicates the increased locomotion at high laser power and replicates the no change in locomotion at low laser power as seen with our constitutive ChR2 experiment. However, the selective dopaminergic axon activation in the PPN is rewarding at both high and low power, while the constitutive ChR2 activation is aversive. We have added these data to supplemental figure S3, and have added text in the results (lines 595624) and discussion (lines 695-734) about this new exciting finding.

While we can’t exclude the possibility of dopamine influence on the electrophysiology experiments (via changes in input resistance or channel properties), the fast synaptic currents measured are uncharacteristic of inhibitory D2 receptor currents (which would be slow), and are inhibited by the GABAa receptor blocker, GABAzine.

(3) Activation of glutamatergic neurons in the caudal PPN elicits locomotion while the same stimulation in the rostral PPN terminates locomotion. In line with this, the authors report important differences in glutamatergic neurons in the rostral vs caudal PPN (Fig. 5). For the behavioral experiments, the location of the optic fiber is not reported. This is essential for the interpretation of the behavioral experiments. Based on the recent literature, inhibiting glutamatergic neurons in the rostral and in the caudal PPN will produce opposing effects.

We absolutely agree the rostral and caudal PPN differences are functionally important. In Figure 7B, we have mapped the location of the optical fiber tip for each experiment. Our implant location was generally in the rostral-middle part of the PPN and we have added this to the methods section of the revised manuscript in lines 887 and 1048. While we did not have many implant locations that were specifically rostral or specifically caudal, we did evaluate the behavioral response for our most rostrally-located implant and our most-caudally located implant in the SN axon stimulation experiment. We found that low-power laser activation of nigral axons in the most rostral implant resulted in increased locomotion but in the most caudal implant resulted in decreased locomotion. This increased locomotion exactly what we would expect when rostral PPN neurons (that normally inhibit movement) are preferentially inhibited, and decreased locomotion is what we would expect when caudal PPN neurons (that normally promote movement) are inhibited. Future experiments using more precise rostral and caudal implant locations will be needed to fully parse out the functional role of rostral vs caudal PPN. See Figure S4 (two green implant sites are circled for one mouse because the implants were bilateral).

(4) Even though the authors made an effort to dissect out the motor component during the RTPP task, this was not entirely achieved. Low laser power was still able to decrease activity following GPe stimulation, causing the animal to spend more time in the stimulated compartment. It is not clear the reason for using RTPP as opposed to CPP, which will not have the confound of the effects on motor activity. The interpretation of these data is problematic.

This is an important consideration, and the reviewer is correct that we can’t completely eliminate a motor contribution to our RTPP experiment. We attempted to minimize potential motor confounds by utilizing unilateral stimulation and our supplemental videos show that the mice can escape the stimulated chamber.

However, to address this comment, we analyzed the second day of RTPP, where no stimulation was applied in either chamber. Specifically, we evaluated the time spent in the stimulated chamber during the first minute of the unstimulated RTPP task. We found that the mice that had SN stimulation still avoided the previously stimulated chamber and the mice that had GPe axon stimulation still preferred the previously stimulated chamber. These data have been added as Figure 7G and in the results section lines 564-575.

(5) The resting membrane potential for cholinergic, glutamatergic, and GABAergic neurons is not reported.

Since a majority of PPN neurons are spontaneously active, we have reported the average membrane voltage during the pre-optical stimulation period in supplementary table 1.

(6) During the RTPP, the animals were stimulated unilaterally with the purpose of reducing the optogenetic effects on locomotion, but no data support this claim. Please report the locomotor measurements during unilateral stimulation.

To address this comment, we have analyzed the speed of the mouse in each compartment (stimulated vs non-stimulated) during the RTPP task. We found that the mean speed does differ, in the direction expected (i.e., mice are on average slower in the GPe stimulated zone where they spend more time, and mice are on average faster in the SNr stimulated zone where they spend less time). This is expected because when the mouse spends more time in a zone, it is more likely to spend time grooming or staying still, but it could still be evidence of motor response to the stimulation. To evaluate how fast the mouse is able to move with and without unilateral stimulation, we measured maximum speed in the stimulated and unstimulated zone. We found that maximum speed does not differ between stimulated and unstimulated zones in either the SNr or GPe group. See Author response image 3.

**Author response image 3. sa4fig3:** 

(7) Given the similarity of the parameters evaluated for all three PPN cell types, the results could be presented in a table, it will be easier to summarize.

This is a good point and we have added supplemental tables 1-4 for key electrophysiological findings.

(8) The text is repetitive in some parts.

We have gone through the results to edit out repetitive text. For example, lines 244-260 and 274-287 have been rewritten for clarity and efficiency.

(9) Lines 609-620: the behavioral effects after SNr stimulation are not mediated by the PPN, please correct.

We have corrected this.

(10) The number of patched GABAergic neurons in the caudal PPN is almost double the number of patched neurons in the rostral PPN. This contrasts with the high density of GABAergic neurons in the rostral PPN reported in the literature, and therefore, the probability of recording GABAergic neurons will be much higher in the rostral PPN. Please comment.

It is true that there are more GABA neurons in the rostral region, but on a sagittal slice, the rostral region occupies a smaller area compared to caudal and there is a notable cluster of GABAergic neurons in the caudal region (Mena-Segovia et al. 2009). The number of visible and healthy cells with obvious fluorescence against background fluorescence in the heavily myelinated tissue of the PPN is unpredictable and it is possible that the dense number of GABA neurons in the rostral region conglomerates the fluorescence of individual cell somas, making it difficult to detect as many rostral neurons. While we did our best to equally patch rostral and caudal neurons based on our best judgment during the experiment, neurons were ultimately designated as ‘rostral’ or ‘caudal’ after post-hoc staining for the cholinergic neurons, as described below.

(11) Describe how the rostral and caudal PPN regions were defined and how the authors ensured consistency across recordings.

We have added more details about the definition of rostral vs caudal PPN in to the methods in lines 1042-1053.

(12) Please report the proportion of GABAergic neurons showing STD vs STP for rostral and caudal PPN. The data in Figure 3 might be averaging out some important differences. Figure 3L suggests some differences in the proportions.

The variability within the GABAergic population was really interesting and we plan to pursue this in the future. We have defined STD as PPR<0.95 and STP as PPR>1.05 and added the proportions of caudal and rostral GABAergic PPN neurons with each type of short-term synaptic plasticity to lines 253-257.

(13) Please report whether the mice’s compartment preferences during the habituation were taken into account for the selection of the laser-on compartment.

Mice were not habituated to the chamber in the unstimulated condition prior to the RTPP experiment. Laser-on side was randomly chosen and counter-balanced between mice. Mice were also randomly assigned to have low laser power RTPP first or high laser power RTPP first. In each case, mice were given an unstimulated 10-minute trial on the day between the first and second RTPP experiment to ‘unlearn’ which side was stimulated and the second RTPP experiment stimulated the opposite chamber compared to the first RTPP experiment. For example, one mouse would have high power stimulation on the striped side on day 1, no stimulation on day 2, and low power stimulation on the spotted side on day 3. This is now explained more thoroughly in lines 564-575 and lines 992-998.

(14) Some references to figure panels are missing in the text.

We have carefully reviewed the manuscript to ensure figure panels are referenced in the text.

(15) The interpretation in lines 724-725 is not supported by the data given that GPe inputs to cholinergic neurons are negligible.

We have reworded much of the discussion.

(16) Some parts of the discussion should go into the “ideas and speculation” subsection of the discussion.

We have rewritten sections of the discussion.

References:

Ferreira-Pinto, Manuel J., Harsh Kanodia, Antonio Falasconi, Markus Sigrist, Maria S. Esposito, and Silvia Arber. 2021. “Functional Diversity for Body Actions in the Mesencephalic Locomotor Region.” Cell 184 (17): 4564-4578.e18. https://doi.org/10.1016/j.cell.2021.07.002.

Goñi-Erro, Haizea, Raghavendra Selvan, Vittorio Caggiano, Roberto Leiras, and Ole Kiehn. 2023. “Pedunculopontine Chx10+ Neurons Control Global Motor Arrest in Mice.” Nature Neuroscience 26 (9): 1516–28. https://doi.org/10.1038/s41593-023-01396-3.

Mena-Segovia, J., B. R. Micklem, R. G. Nair-Roberts, M. A. Ungless, and J. P. Bolam. 2009. “GABAergic Neuron Distribution in the Pedunculopontine Nucleus Defines Functional Subterritories.” The Journal of Comparative Neurology 515 (4): 397–408. https://doi.org/10.1002/cne.22065.

Steinkellner, Thomas, Ji Hoon Yoo, and Thomas S. Hnasko. 2019. “Differential Expression of VGLUT2 in Mouse Mesopontine Cholinergic Neurons.” eNeuro, July. https://doi.org/10.1523/ENEURO.0161-19.2019.

Wang, Hui-Ling, and Marisela Morales. 2009. “Pedunculopontine and Laterodorsal Tegmental Nuclei Contain Distinct Populations of Cholinergic, Glutamatergic and GABAergic Neurons in the Rat.” The European Journal of Neuroscience 29 (2): 340–58. https://doi.org/10.1111/j.1460-9568.2008.06576.x.

Yoo, Ji Hoon, Vivien Zell, Johnathan Wu, Cindy Punta, Nivedita Ramajayam, Xinyi Shen, Lauren Faget, Varoth Lilascharoen, Byung Kook Lim, and Thomas S. Hnasko. 2017. “Activation of Pedunculopontine Glutamate Neurons Is Reinforcing.” The Journal of Neuroscience: The Official Journal of the Society for Neuroscience 37 (1): 38–46. https://doi.org/10.1523/JNEUROSCI.3082-16.2016.